# Robust Spiking Neural Networks Against Adversarial Attacks

**Shuai Wang**[1,3], **Malu Zhang**[1,3],* **Yulin Jiang**[1],**Dehao Zhang**[1], **Ammar Belatreche**[2],
**Yu Liang**[1], **Yimeng Shan**[1], **Zijian Zhou**[1], **Yang Yang**[1], **Haizhou Li**[3,4],

[1]University of Electronic Science and Technology of China [2]Northumbria University,
[3]Shenzhen Loop Area Institute, [4]The Chinese University of Hong Kong, Shenzhen.

## Abstract

Spiking Neural Networks (SNNs) represent a promising paradigm for energy-efficient neuromorphic computing due to their bio-plausible and spike-driven characteristics. However, the robustness of SNNs in complex adversarial environments remains significantly constrained. In this study, we theoretically demonstrate that those threshold-neighboring spiking neurons are the key factors limiting the robustness of directly trained SNNs. We find that these neurons set the upper limits for the maximum potential strength of adversarial attacks and are prone to state-flipping under minor disturbances. To address this challenge, we propose a Threshold Guarding Optimization (TGO) method, which comprises two key aspects. First, we incorporate additional constraints into the loss function to move neurons' membrane potentials away from their thresholds. It increases SNNs' gradient sparsity, thereby reducing the theoretical upper bound of adversarial attacks. Second, we introduce noisy spiking neurons to transition the neuronal firing mechanism from deterministic to probabilistic, decreasing their state-flipping probability due to minor disturbances. Extensive experiments conducted in standard adversarial scenarios prove that our method significantly enhances the robustness of directly trained SNNs. These findings pave the way for advancing more reliable and secure neuromorphic computing in real-world applications.

## 1 Introduction

Spiking Neural Networks (SNNs) (Maass, 1997; Gerstner & Kistler, 2002; Izhikevich, 2003; Masquelier et al., 2008) mimics biological information transmission mechanisms using discrete spikes as the medium for information exchange, representing the cutting edge of neural computation (Cao et al., 2020; Varghese et al., 2016). Spiking neurons fire spikes only upon activation and remain silent otherwise. This event-driven mechanism (Liu & Yue, 2018) promotes sparse synapse operations and avoids multiply-accumulate (MAC) operations, significantly enhancing energy efficiency on neuromorphic platforms (Pei et al., 2019; DeBole et al., 2019; Ma et al., 2024; Pei et al., 2019). Recently, directly training SNNs with surrogate gradient methods (Wu et al., 2018; 2019; Deng et al., 2022; Li et al., 2021; Wang et al., 2025a) has significantly reduced their performance gap with ANNs in classification tasks (Yao et al., 2024a; Shi et al., 2024; Zhou et al., 2024; Wang et al., 2025c; Liang et al., 2025). However, these directly trained SNNs rely on Backpropagation Through Time (BPTT) (Werbos, 1990), thereby inheriting significant robustness issues associated with ANNs.

Directly trained SNNs (Fang et al., 2021b; Zhou et al., 2023; Bu et al., 2022; Duan et al., 2022) using surrogate gradient methods often exhibit a strong dependency on specific patterns or features (Ding et al., 2022; Mukhoty et al., 2024), rendering them particularly sensitive to minor disturbances. This characteristic reduces robustness in complex environments, especially against finely crafted adversarial disturbances (Laskov & Lippmann, 2010). To enhance the robustness of SNNs against adversarial attacks, researchers adapt strategies from ANNs, such as adversarial training (Ho et al., 2022; Ding et al., 2022) and certified training (Zhang et al., 2019; Liang et al., 2022). Furthermore, researchers develop optimization methods tailored to spike-driven mechanisms, integrating with

---

*Corresponding author: maluzhang@uestc.edu.cn

adversarial training to enhance robustness. Some researchers (Sharmin et al., 2020; Ding et al., 2023; El-Allami et al., 2021) utilize the temporal characteristics of SNNs to counteract environmental white noise attacks. Additionally, Evolutionary Leak Factor (Xu et al., 2024), MPD-SGR Jiang et al. (2025) and gradient sparsity regularization (SR) (Liu et al., 2024) significantly enhance the robustness of SNNs against gradient-based attacks. However, a comprehensive and unified analysis of the robustness bottlenecks in directly trained SNNs remains lacking.

In this study, we theoretically demonstrate that threshold-neighboring spiking neurons are a key factor influencing the robustness of directly trained SNNs under adversarial attacks. We find that these neurons provide maximum potential pathways for adversarial attacks and are more prone to state-flipping under minor disturbances. To address this, we propose a Threshold Guarding Optimization (TGO) method. The TGO method aims to: (1) maximize the distance between neurons' membrane potentials and their thresholds to enhance gradient sparsity; (2) minimize the probability of state-flipping in neurons under minor disturbances. A series of experiments in standard adversarial scenarios demonstrates that our TGO method significantly enhances the robustness of directly trained SNNs. The contributions of this work are summarized as follows:

- We theoretically demonstrate that those threshold-neighboring spiking neurons are critical in limiting the robustness of directly trained SNNs under adversarial attacks. These neurons set the upper limits for the maximum potential strength of adversarial attacks and are prone to state-flipping under minor disturbances.
- We propose a Threshold Guarding Optimization (TGO) method, aiming to minimize threshold-neighboring neurons' sensitivity to adversarial attacks. First, we integrate additional constraints into the loss function, distancing the membrane potential from the threshold. Second, we introduce noisy spiking neurons to transit neuronal firing from deterministic to probabilistic, reducing the probability of state flips due to minor disturbances.
- We validate the effectiveness of the TGO method across various adversarial attack scenarios using different training strategies. Extensive experiments demonstrate TGO method achieves state-of-the-art (SOTA) performance in multiple adversarial attacks, significantly enhancing the robustness of SNNs. Notably, TGO method incurs no additional computational overhead during inference, providing a feasible pathway toward robust edge intelligence.

## 2 RELATED WORK

**Spiking Neural Networks:** SNNs offer a promising solution for resource-constrained edge computing (Zhang et al., 2023). To enhance the performance of SNNs, Wu et al. (2018) introduces the spatial-temporal backpropagation (STBP) algorithm, an adaptation of BPTT from Recurrent Neural Networks (RNNs) (Graves & Graves, 2012; Lipton, 2015). This method uses surrogate functions to approximate the non-differentiable Heaviside step function in spiking neurons. Additionally, researchers explore parallel training strategies (Fang et al., 2024) within the ResNet framework (He et al., 2016), shortcut residual connections (Zheng et al., 2021; Hu et al., 2021; Lee et al., 2020; Fang et al., 2021a; Zhang et al., 2025b), and Spike transformer (Li et al., 2022). Notably, SNNs have achieved performance comparable to their ANN counterparts across diverse vision tasks, including image classification (Yao et al., 2024a; Xiao et al., 2025), object detection (Wang et al., 2025b), Event tracking (Shan et al., 2025) and semantic segmentation (Zhang et al., 2025a). Despite surrogate gradient methods (Deng et al., 2023; Yang & Chen, 2023) significantly improve training efficiency, SNNs remain susceptible to adversarial attacks as ANNs (Finlayson et al., 2019; Xu et al., 2020), limiting their applicability in adversarial environments.

**Robustness of SNNs in Adversarial Attacks** While biologically event-driven mechanisms (Marchisio et al., 2020; Hao et al., 2020) enhance SNNs' adaptability in complex environments, empirical studies (Liang et al., 2021; El-Allami et al., 2021) reveal that directly trained SNNs remain vulnerable to adversarial attacks. Initial efforts to mitigate this vulnerability start with adapting Adversarial Training (AT) (Goodfellow et al., 2014; Kundu et al., 2021) and subsequently advance to Regularized Adversarial Training (RAT) (Ding et al., 2022) with Lipschitz analysis. However, these approaches are constrained by additional training overhead and limited portability (Shafahi et al., 2019). Recently, researchers have developed optimization methods tailored to spike-driven mechanisms of SNNs. Such as Hao et al. (2023) enhances intrinsic robustness through rate-temporal information integration. (Xu et al., 2024) introduces FEEL-SNN with random membrane potential decay and innovative

encoding mechanisms, and Ding et al. (2024b) develops gradient SR to strengthen defenses against Fast Gradient Sign Method (FGSM) (Goodfellow et al., 2014) and Projected Gradient Descent (PGD) (Madry, 2017). Despite these advances, these strategies achieve significant enhancements only through synergistic integration with AT and RAT strategies. Moreover, a theoretical analysis of SNNs' inherent vulnerabilities in adversarial environments is still lacking. Thus, devising more effective robustness optimization strategies for SNNs remains a focused research.

## 3 PRELIMINARIES

### 3.1 SURROGATE GRADIENT FOR DIRECT TRAINED SNNS

SNNs effectively model the complex dynamics of biological neurons. Within the Leaky Integrate-and-Fire (LIF) framework, the membrane potential transitions through three key stages: integration, leakage, and firing. During integration, the membrane potential $V[t]$ accumulates over time in response to incoming spikes. When $V[t]$ exceeds a predefined threshold $V_{\text{th}}$, it triggers a spike that may influence downstream neurons. Following the spike, the membrane potential is reset to a specified baseline $V_{\text{reset}}$, preparing the neuron for subsequent inputs, which can be described as:

$$V[t] = \tau U[t-1] + WS[t], \tag{1}$$

$$S[t] = \Theta\left(V[t] - V_{\text{th}}\right), \tag{2}$$

$$U[t] = V[t]\left(1 - S[t]\right) + V_{\text{reset}}S[t], \tag{3}$$

where $\tau$ is the membrane time constant, $W$ represents the synaptic weights, $S[t]$ denotes the spike at time $t$, and $\Theta(\cdot)$ is the Heaviside step function, indicating firing when $V[t]$ exceeds $V_{\text{th}}$. In the directly trained SNNs, the total loss $L$ with respect to the weights $W$ can be described as:

$$\frac{\partial L}{\partial W} = \sum_t \frac{\partial L}{\partial S[t]} \frac{\partial S[t]}{\partial V[t]} \frac{\partial V[t]}{\partial W}. \tag{4}$$

where $\frac{\partial S[t]}{\partial V[t]}$ represents the gradient of a non-differentiable step function involving the derivative of the Dirac $\delta$-function, which is typically replaced by surrogate gradients with derivable curves. Various forms of surrogate gradients have been utilized, such as rectangular (Wu et al., 2018; 2019), triangular (Esser et al., 2016; Rathi & Roy, 2020), and exponential (Shrestha & Orchard, 2018) curves. Surrogate gradients provide a differentiable approximation to non-differentiable functions.

### 3.2 ADVERSARIAL ATTACKS

Adversarial attacks, including the Fast Gradient Sign Method (FGSM) (Goodfellow et al., 2014)and Projected Gradient Descent (PGD) (Madry, 2017), rely on the model's gradient information to craft adversarial examples. FGSM generates such examples by applying a single-step perturbation designed to maximize the model's prediction error. The adversarial input is calculated as:

$$\mathbf{x}_{\text{adv}} = x + \epsilon \cdot \text{sign}(\nabla_x \mathcal{L}(x, y_{\text{true}})), \tag{5}$$

where $\mathbf{x}_{\text{adv}}$ is the adversarial example, $x$ is the original input, $\epsilon$ is the perturbation magnitude, $\mathcal{L}(x, y_{\text{true}})$ is the loss function, and $\text{sign}(\nabla_x \mathcal{L}(x, y_{\text{true}}))$ gives the sign of the gradient concerning the input. This process leverages the model's loss landscape to introduce minimal disturbances that significantly increase the classification error. Building on this, PGD iteratively refines adversarial examples by applying gradient updates and projecting them back into a bounded $\epsilon$-ball centered on the original input. The attack rule of PGD can be expressed as:

$$\mathbf{x}_{\text{adv}}^{t+1} = \text{Clip}_{x,\epsilon}\left(\mathbf{x}_{\text{adv}}^t + \alpha \cdot \text{sign}(\nabla_x \mathcal{L}(\mathbf{x}_{\text{adv}}^t, y_{\text{true}}))\right), \tag{6}$$

where $\mathbf{x}_{\text{adv}}^t$ is the adversarial example at iteration $t$, $\alpha$ is the step size, and $\text{Clip}_{\mathbf{x},\epsilon}$ ensures that the perturbation remains within the prescribed $\epsilon$-ball. PGD employs a multi-step approach to more precisely explore the disturbance space, producing adversarial examples closer to the optimal solution. At the same time, it strictly constrains the magnitude of disturbances, ensuring the perturbed input remains nearly indistinguishable from the original data to human observers. Multi-PGD improves robustness evaluation by reducing sensitivity to initialization and increasing the likelihood of finding stronger adversarial examples. APGD (Auto-PGD) automates and stabilizes the attack by adaptively adjusting hyperparameters such as step size and scheduling, enabling consistently strong performance with minimal manual tuning. These two methods serve as standard benchmarks for evaluating the adversarial defense capabilities of neural networks.

## 4 METHODS

### 4.1 ROBUSTNESS ANALYSIS OF DIRECTLY TRAINED SNNS

To explore the key factors affecting the adversarial robustness of SNNs, we conduct a detailed analysis of the SNNs' dynamic properties under adversarial attacks. Our findings highlight two critical vulnerabilities associated with those threshold-neighboring spiking neurons. First, they establish a theoretical upper bound for the maximum potential strength of adversarial attacks. Second, they exhibit a higher probability of state-flipping under minor disturbances.

**maximum potential gradient-based attack path:** Adversarial attacks strategically modify input disturbances to maximize the expected loss, with these disturbances typically aligning with the gradient of the input data. The metric $\mathcal{R}_{\text{adv}}(f, x, \epsilon)$ quantifies the maximum potential strength of an adversarial attack on the neural network $f$ at a specific input $x$, where the dis-

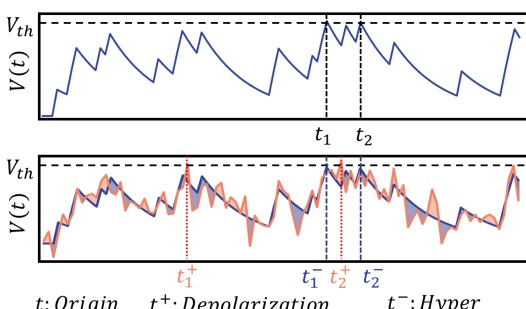

Figure 1: Red traces represent membrane potential dynamics of spiking neurons under adversarial attack. Only membrane potentials near thresholds undergo spike pattern transitions, while others remain unchanged.

turbances are constrained within a unit $\ell_p$-norm ball and scaled by the factor $\epsilon$. This measure is mathematically expressed as follows:

$$\mathcal{R}_{\text{adv}}(f, x, \epsilon) = \max_{\|\delta\|_p \leq 1} \|f(x + \epsilon\delta) - f(x)\|_2^2. \tag{7}$$

Eq.7 aims to find the disturbance $\delta$ that maximizes the squared output difference while remaining within an $\ell_p$-norm ball. Applying Taylor expansion with the Lagrange remainder, we expand it as follows:

$$f(x + \epsilon\delta) = f(x) + J_f(x)(\epsilon\delta) + \frac{(\epsilon\delta)^2}{2}H_f(x + \xi\epsilon\delta). \tag{8}$$

where $\xi \in (0, 1)$ serves as the expansion coefficient. Let $f : \mathbb{R}^n \to \mathbb{R}^m$ be a continuously differentiable neural network function at point $x$, and let $\epsilon > 0$ be sufficiently small. The matrix $H_f(x)$ and $J_f(x)$ respectively denote the Hessian and Jacobian matrix of the function $f(\cdot)$ at point $x$.

Utilizing the Cauchy-Schwarz inequality Steele (2004) and assuming that $\|\delta\|_p \leq 1$, the upper bound of the difference caused by the minor disturbance can be expressed as:

$$\|f(x + \epsilon\delta) - f(x)\|_2^2 \leq \left(\epsilon\|J_f(x)\|_2 + \frac{\epsilon^2}{2}\lambda_{\text{Hmax}}\right)^2, \tag{9}$$

Eq.9 represents the sensitivity of $f$ at $x$ to disturbances along $\delta$. $\|J_f(x)\|$ is the $\ell_2$ norm of the Jacobian matrix, and $\lambda_{\text{Hmax}}$ is the maximum eigenvalue of $H_f(x)$. Then, we derive the upper bound on the $\mathcal{R}_{\text{adv}}$:

$$\mathcal{R}_{\text{adv}}(f, x, \epsilon) \leq \epsilon^2\|J_f(x)\|_2^2 + O(\epsilon^2). \tag{10}$$

The Jacobian matrix $J_f(x)$ can be expressed as the collection of gradients of each component of the function:

$$\|J_f(x)\|_2^2 = \lambda_{\text{Jmax}}\left(\sum_{i=1}^m \nabla f_i(x)\nabla f_i(x)^T\right). \tag{11}$$

According to Eq.11, the sensitivity of SNNs to adversarial disturbances is correlated with the $\ell_2$ norm of their Jacobian matrix, where higher gradient $\ell_2$ norms indicate greater susceptibility to adversarial attacks. Notably, directly trained SNNs typically rely on surrogate gradients, which exhibit peak values near the threshold. As the number of threshold-neighboring spiking neurons increases, the $\ell_2$ norm of the gradients in SNNs also rises, thereby enlarging $\mathcal{R}_{\text{adv}}(f, x, \epsilon)$. Consequently, these neurons significantly raise the theoretical upper limit of adversarial perturbation strength. Details can be found in Appendix B.

**Strong State-flipping Probability**: Adversarial attacks introduce carefully crafted small disturbances into the input data, achieving their disruptive effects. These disturbances propagate through the multi-layers, causing state-flipping in spiking neurons and ultimately altering the final output. Due to the spike-driven nature, changes occur only when spiking neurons' membrane potential crosses the threshold.

**Theorem 1** *Let $V[t]$ be the membrane potential, $V_{\text{th}}$ the threshold, and $\eta[t] \sim \mathcal{N}(0, \sigma^2)$ random perturbation. The probability $P_{\text{flip}}$ of each neuron's flipping is given by:*

$$P_{\text{flip}} = \begin{cases} \Phi\left(\frac{V_{\text{th}} - V[t]}{\sigma}\right), & \text{if } V[t] \geq V_{\text{th}}, \\ 1 - \Phi\left(\frac{V_{\text{th}} - V[t]}{\sigma}\right), & \text{if } V[t] < V_{\text{th}}. \end{cases}$$

*where $\Phi$ denotes the cumulative distribution function (CDF) of the standard normal distribution.*

Theorem 1 defines the relationship between neuronal membrane potential and their state-flipping probability. Specifically, when $V[t] \geq V_{\text{th}}$, the neuron output switches from 1 to 0, and when $V[t] < V_{\text{th}}$, it flips from 0 to 1. Since the CDF of the standard normal distribution $\Phi(\cdot)$ is increasing monotonically, $P_{\text{flip}}$ increases as the membrane potential $V[t]$ approaches the threshold potential $V_{\text{th}}$, whether $V[t]$ is above or below $V_{\text{th}}$. As shown in Fig. 1, small noise perturbations mainly cause state flips in neurons near the threshold, while fluctuations elsewhere have little effect on spike outputs. Therefore, state flipping directly increases the instability of activation patterns, and its impact on the upper bound of adversarial attacks is summarized as follows.

**Theorem 2** *For a discrete spike pattern mapping $f : \mathbb{R}^n \to \mathbb{R}^m$, small perturbations $\varepsilon\delta$ around input $x$ induce a finite set of activation pattern transitions. The adversarial robustness upper bound can be approximated as:*

$$R_{adv}(f, x, \varepsilon) \leq \varepsilon^2 \max_{1 \leq k \leq K} \|A_{\mathcal{A}_k}\|_{p \to 2}^2,$$

*where $K$ denotes the number of activation regions intersecting the perturbation ball $B_\varepsilon(x)$, and $A_{\mathcal{A}_k} \in \mathbb{R}^{m \times n}$ is the affine transformation matrix for activation pattern $\mathcal{A}_k = \{(l, i) : u_i(x) \geq \theta_i\}$.*

Theorem 2 proves that a larger $K$ expands the set of possible transformations (represented by affine matrices $A_{\mathcal{A}_k}$), thereby increasing the potential impact of adversarial perturbations on the system. As the proportion of threshold-near neurons increases, $K$ grows, leading to heightened adversarial vulnerability. In summary, threshold-neighboring spiking neurons play a crucial role in the adversarial robustness of SNNs. To address this, we propose an optimization strategy designed to mitigate their impact, thereby strengthening the overall resilience of SNNs in adversarial environments.

## 4.2 THRESHOLD GUARDING OPTIMIZATION METHOD

### 4.2.1 MEMBRANE POTENTIAL CONSTRAINTS

The surrogate gradients of threshold-neighboring spiking neuron, significantly influence the $\|J_f(x)\|_2^2$ of SNNs. To mitigate this effect, we propose additional constraints at each spiking neuron layer to optimize the membrane potential distribution, ensuring it remains as distant as possible from the threshold. The membrane potential constraint function can be described as:

$$\mathcal{C}(V(t)_l) = \frac{1}{TN} \sum_{i=1}^{n} \max(0, \delta - |V(t)_i - V_{\text{th}}|), \tag{12}$$

$\mathcal{C}(V(t)_l)$ computes the average quadratic penalty when membrane potentials $V(t)_i$ of neurons in layer $l$ approach the firing threshold $V_{\text{th}}$. $N$ and $T$ represent the number of time steps and the total number of layers in the SNNs, respectively. The hyperparameter $\delta$ establishes a margin around $V$th where proximate potentials incur proportional penalties. Subsequently, we integrate this constraint with the target loss function, defining the overall loss within the framework of Lagrangian constraints (Kim & Jeong, 2021; Yoo & Jeong, 2023), which can be expressed as:

$$\mathcal{L}(\mathbf{x}, \lambda) = \mathcal{L}oss(\mathbf{x}) + \lambda \sum_{l} \mathcal{C}(V(t)_l). \tag{13}$$

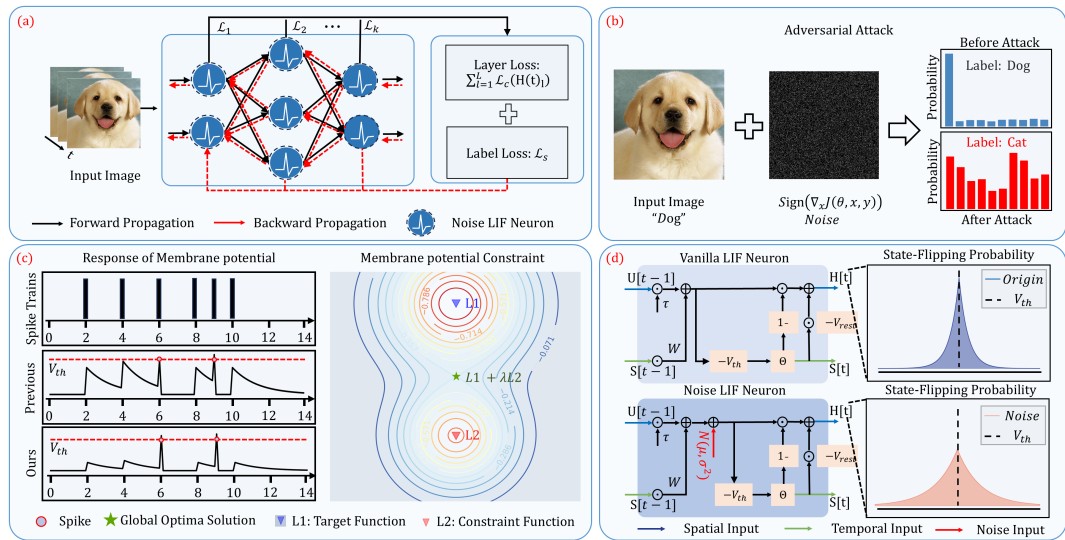

Figure 2: Mechanism and working principle of the TGO method. (a) The TGO method combines membrane potential constraints with noisy LIF neuron models for adversarial defense. (b) Gradient-based adversarial attacks illustrate how disturbances affect input images. (c) The joint optimization of the objective and constraint functions drives neuron membrane potentials away from the firing threshold. (d) The noisy LIF model effectively reduces the probability of state flips caused by small input disturbances, enhancing model stability.

Here, $\mathcal{L}oss(\mathbf{x})$ is the original loss function, $\mathcal{C}(V(t)_l)$ represents the penalty term for the membrane potentials across all layers, and $\lambda$ is a dynamically adjusted parameter that controls the significance of the constraint. We reveal that using a fixed magnitude for $\lambda$ hinders network convergence and constraint satisfaction. Specifically, a larger $\lambda$ leads to significant performance degradation and poor convergence during the initial training phase, while a smaller $\lambda$ fails to enforce the constraint effectively. Therefore, to achieve an optimal balance between gradients sparsity and performance, we propose dynamic $\lambda$, which can be described as:

$$\lambda = 0.5 \times \lambda_{\max} \times \left( 1 - \cos \left( \frac{\pi \times \text{epoch}}{\text{epoch}_{max}} \right) \right).$$  (14)

In the dynamic adjustment strategy, $\lambda$ is initially set low to allow extensive exploration of parameter space and prevent premature constraints on $V[t]$ distributions. As $\lambda$ increases, constraints intensify, pushing $V[t]$ further from $V_{\text{th}}$ and ensuring strict adherence to the operational threshold. Membrane potential constraints effectively reduce the number of threshold-neighboring neurons, thereby decreasing $|J_f(x)|_2^2$ of SNNs. However, during training, some neurons inevitably remain near the threshold due to their significant impact on the loss function. Therefore, additional strategies are required to enhance the robustness of these critical neurons.

### 4.2.2 Noisy Spike Neurons for Mitigating State-flip Probability

As previously mentioned, these neurons exhibit high sensitivity to minor disturbances, readily undergoing state flipping by crossing the firing threshold. Such flipping not only increases output instability but can also severely disrupt the network's overall output through cascade effects. Consequently, we introduce the Noisy-LIF neuron model (Gerstner et al., 2014) as a complementary mechanism to membrane potential constraints, significantly reducing the probability of state flipping in critical neurons and thereby enhancing the robustness of directly trained SNNs against adversarial attacks. The dynamics of Noisy-LIF can be described as:

$$V[t] = \tau U[t-1] + WS[t] + \xi[t],$$  (15)

where $\xi[t]$ denotes Gaussian white noise, $V[t] = \tau U[t-1] + WS[t]$ is the membrane potential of the LIF model before the addition of noise. In the traditional LIF model, spike generation is

deterministic: neurons emit a spike whenever the membrane potential $V[t]$ surpasses the firing threshold $V_{\text{th}}$. Details are described in Appidex.E. As a result, even small noise perturbations can cause significant output flips if they drive the membrane potential above the threshold. By introducing $\xi[t]$, the output transitions to an expected value rather than a binary outcome. For small changes $\Delta V[t]$ around $V_{\text{th}}$, the change in spike probability, interpreted as the flipping probability, can be approximated using a Taylor expansion with the first order:

$$\Delta P(S_l = 1 \mid V[t], \xi[t]) \approx \frac{1}{\sigma} \phi(z) \, \Delta V[t], \tag{16}$$

where $z = \frac{V_{\text{th}} - V[t] - \mu}{\sigma}$ and $\phi(\cdot)$ represents the PDF of the standard normal distribution. Then the derivative of the approximated flipping probability $\Delta P(S_l = 1 \mid V[t], \xi[t])$ with respect to $\sigma$, denoted as $\frac{\partial(\Delta P)}{\partial \sigma}$ can be expressed as:

$$\frac{\partial(\Delta P)}{\partial \sigma} = \Delta V[t] \frac{\phi(z)}{\sigma^2} \left( z^2 - 1 \right). \tag{17}$$

For values of $V[t]$ close to $V_{\text{th}}$, an appropriate choice of $\sigma$ can ensure that $z^2 < 1$. Under these conditions, $\frac{\partial(\Delta P)}{\partial \sigma}$ is negative, implying that the flipping probability decreases monotonically with increasing $\sigma$. This observation indicates that increasing the noise level $\sigma$ reduces the sensitivity of the flipping probability to small perturbations in the membrane potential $V[t]$, thereby enhancing the buffering effect of the Noisy-LIF neuron against such disturbances. Overall, the membrane potential constraints and noisy-LIF neurons in the TGO method work synergistically. The constraints ensure that most neuronal potentials are distanced from the threshold, while the noisy-LIF neurons further reduce the probabilities of state flipping in those threshold-neighboring spiking neurons. This integrated approach significantly enhances the adversarial robustness of SNNs.

Table 1: Comparative robustness performance under different training strategies on CIFAR-10 using WRN-16. SOTA performances are marked in  gray .

| Model | Train | Method | Clean | FGSM | RFGSM | PGD7 | PGD10 | PGD20 | PGD40 |
|-------|-------|--------|-------|------|-------|------|-------|-------|-------|
| | | | | | Adversarial Attacks | | | | |
| WRN-16 | BPTT | Vanilla Deng et al. (2022) | 93.32 | 14.05 | 31.21 | 0.00 | 0.00 | 0.00 | 0.00 |
| | | TGO(Ours) | 88.79 | 51.40 | 71.38 | 11.67 | 6.14 | 1.52 | 0.45 |
| | AT | AT Kundu et al. (2021) | 91.32 | 39.14 | 74.31 | 21.15 | 17.45 | 14.41 | 12.93 |
| | | TGO(Ours) | 88.16 | 63.03 | 79.69 | 42.52 | 35.01 | 24.76 | 20.11 |
| | RAT | RAT Ding et al. (2022) | 91.44 | 42.02 | 75.89 | 23.90 | 19.81 | 16.24 | 14.18 |
| | | TGO(Ours) | 87.33 | 69.16 | 79.28 | 54.59 | 47.69 | 38.07 | 33.13 |

## 5 EXPERIMENTS

### 5.1 EXPERIMENTS SETTING

In this study, the TGO method is evaluated in image classification tasks using the CIFAR-10, CIFAR-100 (Krizhevsky et al., 2009) datasets. The employed network architectures include VGG-11 and WideResNet-16 with a width of 4 and depth of 16 (Xu et al., 2024; Ding et al., 2024a;b). All SNN models are simulated for $T=4$ time steps. Unless otherwise specified, the default $\lambda_{\max}$ is set to 0.4 for WRN-16 and 0.6 for VGG-11. We employ multiple adversarial attack methods, including FGSM (Goodfellow et al., 2014) and RFGSM (Wong et al., 2020) and PGD (Madry, 2017). They are all assessed using a fixed attack intensity of 8/255 and a step size of 0.01, with the number of iterations specified in the attack name (e.g., PGD 10). Notably, RFGSM refers to introducing a small random disturbance samples before applying FGSM. Moreover, three training strategies are implemented to test our TGO method: first, a vanilla training strategy (BPTT) Wu et al. (2018) using original images, which incurs no additional training costs; second, AT Kundu et al. (2021) strategy involves training with white-box PGD attacks (attack intensity of 2/255 and step size $k = 2$); third, RAT incorporates Lipschitz penalties Ding et al. (2022) into the adversarial training.

Table 2: Comparative results of robustness across different methods and conditions on cifar100. SOTA performances are marked in ⬛ gray .

| Model | Train | Method | Adversarial Attacks | | | | | | |
|---|---|---|---|---|---|---|---|---|---|
| | | | Clean | FGSM | RFGSM | PGD7 | PGD10 | PGD20 | PGD40 |
| VGG-11 | BPTT | Vanilla Deng et al. (2022) | 71.42 | 5.92 | 24.95 | 0.18 | 0.07 | 0.04 | 0.02 |
| | | DLIF Ding et al. (2024a) | 70.79 | 6.95 | - | 0.08 | 0.05 | 0.00 | 0.00 |
| | | StoG Ding et al. (2024b) | 70.44 | 8.27 | - | 0.49 | - | - | - |
| | | TGO(Ours) | 69.47 | 17.20 | 38.23 | 2.30 | 1.33 | 0.69 | 0.42 |
| | AT | AT Kundu et al. (2021) | 66.27 | 17.20 | 45.13 | 8.30 | 9.62 | 8.16 | 7.52 |
| | | StoG Ding et al. (2024b) | 66.37 | 23.45 | - | 14.42 | - | - | - |
| | | TGO(Ours) | 65.93 | 24.16 | 47.90 | 12.83 | 10.12 | 8.72 | 6.59 |
| | RAT | RAT Ding et al. (2022) | 67.76 | 20.87 | 46.21 | 11.14 | 9.34 | 7.66 | 6.90 |
| | | StoG Ding et al. (2024b) | 62.26 | 33.40 | - | 23.15 | - | - | - |
| | | TGO(Ours) | 65.64 | 33.84 | 51.44 | 18.84 | 14.59 | 10.57 | 8.90 |
| WRN-16 | BPTT | Vanilla Deng et al. (2022) | 73.46 | 6.36 | 11.15 | 0.01 | 0.00 | 0.00 | 0.00 |
| | | DLIF Ding et al. (2024a) | 73.85 | 8.08 | - | 0.00 | 0.00 | 0.00 | 0.00 |
| | | SR Ding et al. (2024b) | 69.15 | 9.84 | 34.43 | 1.22 | 0.86 | 0.52 | 0.3 |
| | | TGO(Ours) | 69.04 | 23.90 | 41.17 | 3.26 | 1.84 | 0.67 | 0.35 |
| | AT | AT Kundu et al. (2021) | 68.57 | 21.18 | 46.60 | 11.14 | 9.22 | 7.50 | 6.72 |
| | | DLIF Ding et al. (2024a) | 65.86 | 25.90 | - | 15.20 | 14.03 | 13.37 | 13.30 |
| | | SR Ding et al. (2024b) | 62.8 | 28.27 | 48.91 | 21.93 | 18.46 | 12.28 | 9.78 |
| | | TGO(Ours) | 64.49 | 41.99 | 55.01 | 27.78 | 22.75 | 15.99 | 12.91 |
| | RAT | RAT Ding et al. (2022) | 67.59 | 25.07 | 48.92 | 13.60 | 11.41 | 9.07 | 8.35 |
| | | DLIF Ding et al. (2024a) | 66.57 | 33.05 | - | 18.75 | 16.23 | 12.16 | 9.44 |
| | | TGO(Ours) | 64.22 | 40.35 | 54.53 | 25.79 | 20.93 | 14.02 | 10.77 |

## 5.2 COMPARE WITH THE SOTA METHODS

To evaluate the effectiveness of the proposed TGO method, we implement three training strategies (BPTT, AT, and RAT) and compare them with other SOTA robustness methods. Specifically, we replicate the AT and RAT configurations. Table.1 and Table.2 reports the classification accuracy of various network architectures under different attack scenarios on the CIFAR-10 and CIFAR-100 datasets. Our results show that the TGO method consistently achieves SOTA performance across nearly all tested architectures. Significantly, under the BPTT training strategy, our TGO method enhances performance by 10-20% in FGSM and RFGSM attack scenarios. It outperforms other robustness methods.

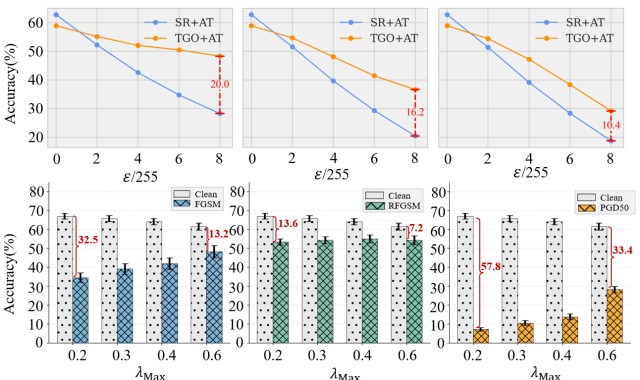

Figure 3: Performance comparison of TGO (ours) and SR with AT across different perturbation budgets $\epsilon$ and different $\lambda_{\max}$. Experiments are conducted on CIFAR-100 dataset using WRN-16 architecture.

Additionally, the TGO strategy is highly compatible with both AT and RAT approaches, achieving approximately 10% performance improvement under PGD10, PGD20, and PGD40 attacks in the WRN-16 model. Similar to other constraint-based approaches, our method achieving a 3%-5% increase in performance under adversarial attacks. We conducted experiments across various per-

turbation magnitudes $\epsilon$ (2, 4, 6, 8/255) and different $\lambda_{\max}$ values (0.2, 0.3, 0.4, 0.6). As shown in Fig. 4, TGO consistently outperforms SR across all perturbation levels, demonstrating its superior robustness. Additionally, increasing $\lambda_{\max}$ results in a trade-off: while clean accuracy decreases, robustness improves, highlighting the balance between accuracy and robustness.

Moreover, we evaluated the TGO method against more advanced attack strategies: Auto-PGD (APGD) (Croce & Hein, 2020) and Multi-Targeted PGD (MTPGD) (Gowal et al., 2019). MTPGD presents unique challenges due to its multi-targeted nature, simultaneously considering multiple misclassification objectives. We implemented MTPGD with random initialization to avoid gradient masking and gradient clipping for stable optimization. We take comparative analysis of our TGO method and the SR method, using consistent hyperparameter settings (momentum=0.9, $\epsilon$=8/255). The incorporation of noisy-LIF in TGO introduces randomness, thereby enhancing its adversarial robustness with inherent variability. We further evaluate TGO's performance under the EoT constraint Athalye et al. (2018). As shown in Table 3, TGO outperforms SR under both MTPGD and APGD attacks even with the EoT constraint. Results across all attack scenarios conclusively demonstrate TGO's significant robustness improvements against diverse adversarial threats.

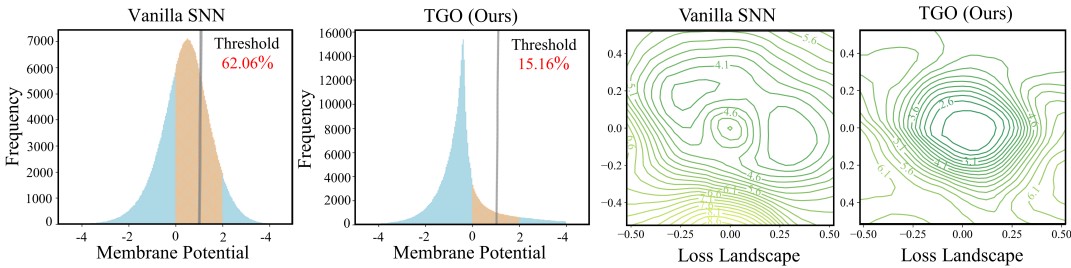

Figure 4: Comparison of membrane potential distributions and loss landscapes: The TGO-optimized SNN decreases membrane potentials near the threshold by approximately 40% and effectively circumvents adversarial traps during RFGSM attacks.

## 5.3 ABLATION STUDY

In this study, we evaluate the two core components of the TGO method: membrane potential constraint (MC) and the noisy-LIF (NLIF) model through a series of ablation experiments. The experiments are performed using two training strategies: vanilla BPTT and RAT, with the CIFAR-100 dataset and VGG-11 architecture ensuring result reliability.

As shown in Table.4, our experimental results reveal several key findings. First, the vanilla SNN exhibits significant vulnerability to FGSM attacks, achieving only 5.92% classification accuracy, highlighting the urgent need to improve its adversarial robustness. Second, each individual component of TGO significantly improves network performance in adversarial attacks, confirming their efficacy in strengthening the robustness of SNNs. Notably, MC and NLIF enhance each other synergistically rather than functioning independently, which further confirms TGO is a holistic protection strategy specifically targeting neurons near the threshold.

Table 3: Performance on MTPGD (Gowal et al., 2019) and APGD (Croce & Hein, 2020) with EoT Athalye et al. (2018) on CIFAR-100 (WRN-16); APGD uses CE loss; EoT repeats $N$=3.

| MTPGD Attack | | | | |
|---|---|---|---|---|
| **Model** | **7** | **10** | **20** | **40** |
| AT Kundu et al. (2021) | 10.01 | 7.56 | 5.66 | 3.92 |
| SR+AT Ding et al. (2024b) | 16.88 | 13.67 | 9.52 | 7.33 |
| TGO+AT(EoT) | **21.23** | **16.35** | **10.58** | **7.40** |
| APGD Attack | | | | |
| **Model** | **7** | **10** | **20** | **40** |
| AT Kundu et al. (2021) | 9.34 | 6.85 | 4.34 | 3.62 |
| SR+AT Ding et al. (2024b) | 14.48 | 12.78 | 8.83 | 7.20 |
| TGO+AT(EoT) | **18.93** | **14.32** | **9.71** | **7.53** |

To better understand TGO's regulatory mechanism on the dynamics of spiking neurons, we analyze the membrane potential distributions of both vanilla and TGO-optimized SNNs. As shown in

Table 4: Ablation study of our TGO method on CIFAR-100 with VGG-11.

| | | BPTT | | | | RAT | | | |
|---|---|---|---|---|---|---|---|---|---|
| MC | NLIF | Clean | FGSM | RFGSM | PGD40 | Clean | FGSM | RFGSM | PGD40 |
| ✗ | ✗ | 71.4 | 5.9 | 25.0 | 0.0 | 67.8 | 20.9 | 46.2 | 6.9 |
| ✓ | ✗ | 64.3 (-7.2) | 17.1 (+11.2) | 25.9 (+0.9) | 0.5 (+0.5) | 61.4 (-6.4) | 26.2 (+5.3) | 42.7 (-3.5) | 6.2 (-0.7) |
| ✗ | ✓ | 70.6 (-0.8) | 8.1 (+2.1) | 31.7 (+6.6) | 0.1 (+0.1) | 68.1 (+0.4) | 25.2 (+4.3) | 50.1 (+3.9) | 9.1 (+2.2) |
| ✓ | ✓ | 66.9 (-4.6) | 21.5 (+15.5) | 39.1 (+14.1) | 0.5 (+0.5) | 63.3 (-3.8) | 33.8 (+13.0) | 50.8 (+4.6) | 9.3 (+2.4) |

Fig.4, TGO reduces the number of threshold-neighboring neurons by 40%, strongly supporting our hypothesis that threshold-neighboring neurons are key to adversarial robustness in SNNs.

Additionally, we compare the loss landscapes of vanilla and TGO-optimized SNNs under RFGSM attacks with the BPTT training strategy. The loss landscape of our TGO method exhibits smoother gradient trajectories, while the vanilla SNNs show multiple local optima and peaks, demonstrating the effectiveness of TGO in defending against gradient-based adversarial attacks. Moreover, we present heatmaps of $\nabla_x f_y$ for CIFAR-100 examples Tsipras et al. (2018); Liu et al. (2024). As shown in Fig.5, the TGO method produces a sparser gradient, with heatmap contours more closely to the original image than the vanilla SNNs.

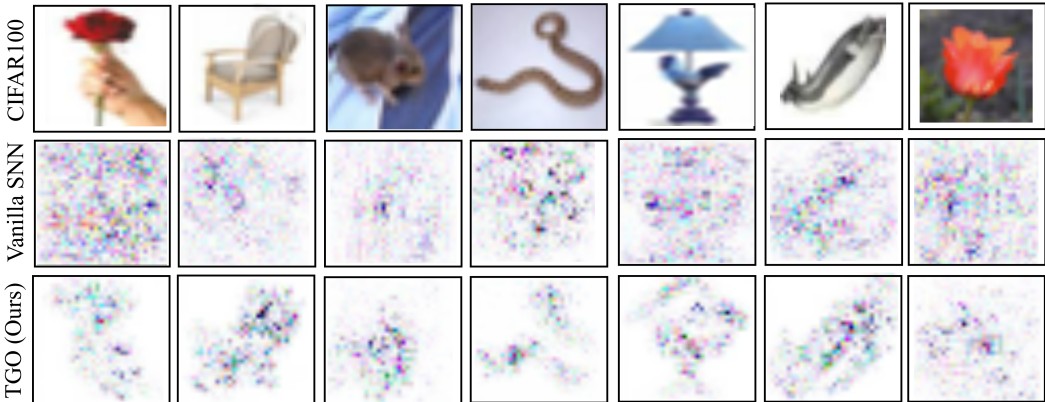

Figure 5: Heatmaps of $\nabla_x f_y$, where $f$ denotes a vanilla SNN or our TGO-optimized SNN.

## 6 CONCLUSION

This study thoroughly analyzes the vulnerabilities of directly trained SNNs under adversarial attack conditions and theoretically confirms that threshold-neighboring spiking neurons define the upper limits of adversarial attack effectiveness. To address this issue, we propose a TGO method, which consists of two aspects. First, membrane potential constraints distance neurons from their thresholds, thereby reducing the upper limits of adversarial attacks. Second, noisy-LIF model transitions the neuronal firing mechanism from deterministic to probabilistic, effectively reducing the probability of state flips caused by minor disturbances. Extensive experiments prove that our TGO method significantly enhances the robustness of directly trained SNNs against various adversarial attacks.

## 7 ACKNOWLEDGMENTS

This work was supported by the National Natural Science Foundation of China (Grants 62576080 and 62220106008), the Sichuan Science and Technology Program (Grant 2024NSFTD0034), the Guangdong Introducing Innovative and Entrepreneurial Teams (Grant 2023ZT10×044), and the Shenzhen Science and Technology Research Fund (Grant JCYJ20220818103001002).

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

## A    ROBUSTNESS OF OUR TGO METHOD FOR NEUROMORPHIC DATASETS

Table 5 illustrates the performance comparison between our proposed TGO+AT method and the standard Adversarial Training (AT) baseline with WideResNet-16 on the DVS-CRIAF10 neuromorphic dataset. In this study, we integrated TGO with conventional adversarial training, resulting in consistent improvements across all evaluation metrics. As evidenced in the table.5, TGO+AT not only enhances clean sample accuracy by 1.9% but also significantly improves robustness against various adversarial attacks, with particularly notable gains against stronger iterative attacks such as PGD7 (+4.1%) and PGD30 (+3.6%). Furthermore, we evaluated the efficacy of our approach

Table 5: Performance Comparison on DVS-CRIAF10 Dataset

| Model | Clean | FGSM | RFGSM | PGD7 | PGD10 | PGD30 | PGD50 |
|---|---|---|---|---|---|---|---|
| AT | 72.2 | 57.4 | 65.2 | 46.9 | 45.6 | 45.2 | 44.2 |
| **TGO+AT** | **74.1** | **59.7** | **68.0** | **51.0** | **48.1** | **48.8** | **46.2** |

against event-based attacks (Yao et al., 2024b) using WideResNet-16 on DVS-CRIAF10 (Li et al., 2017). The Attack Success Rate (ASR) for standard AT was measured at 44.32%, whereas AT+TGO achieved a substantially lower ASR of 31.0%. This reduction in ASR indicates enhanced robustness, further demonstrating the effectiveness of the TGO methodology in mitigating event-based adversarial attacks.

## B    GRADIENT-BASED UPPER BOUND OF ADVERSARIAL ATTACK

**Theorem**: Let $f : \mathbb{R}^n \to \mathbb{R}^m$ be a continuously differentiable neural network function at point $x$, and let $\epsilon > 0$ be sufficiently small. The adversarial perturbation measure $\mathcal{R}_{\text{adv}}(f, x, \epsilon)$ satisfies:

$$\mathcal{R}_{\text{adv}}(f, x, \epsilon) \leq \epsilon^2 \|J_f(x)\|_2^2 + O(\epsilon^3), \tag{18}$$

where $J_f(x)$ is the Jacobian matrix of function $f(\cdot)$ at point $x$.

**Proof**: We begin by defining our objective as the maximization of the squared difference between $f(x + \epsilon\delta)$ and $f(x)$, subject to the constraint $\|\delta\|_p \leq 1$:

$$\mathcal{R}_{\text{adv}}(f, x, \epsilon) = \underset{\delta}{\arg\max} \left\{ \|f(x + \epsilon\delta) - f(x)\|_2^2 : |\delta|_p \leq 1 \right\}. \tag{19}$$

Next, we apply Taylor's theorem with the Lagrange remainder for the vector-valued function $f(x+\epsilon\delta)$ around point $x$. Specifically, for $\delta$ with $\|\delta\|_p \leq 1$, there exists $\xi \in (0, 1)$ such that:

$$f(x + \epsilon\delta) = f(x) + J_f(x)(\epsilon\delta) + \frac{(\epsilon\delta)^2}{2} H_f(x + \xi\epsilon\delta), \tag{20}$$

where $J_f(x)$ is the Jacobian matrix of the function $f(\cdot)$ at the point $x$, and $H_f$ is the Hessian tensor of second derivatives. Substituting this expansion into the squared difference, we obtain:

$$\|f(x + \epsilon\delta) - f(x)\|_2^2 = \left\| J_f(x)(\epsilon\delta) + \frac{(\epsilon\delta)^2}{2} H_f(x + \xi\epsilon\delta) \right\|_2^2 \tag{21}$$

$$\leq \left( \|J_f(x)(\epsilon\delta)\|_2 + \frac{1}{2}\|(\epsilon\delta)^2 H_f(x + \xi\epsilon\delta)\|_2 \right)^2. \tag{22}$$

Combining Eq. 19 and the Cauchy-Schwarz inequality, we can systematically expand each component of the Eq. 22. For the first term, it can be expand as follows:

$$\|J_f(x)(\epsilon\delta)\|_2 \leq \|J_f(x)\|_2 \|\epsilon\delta\|_2 \leq \epsilon\|J_f(x)\|_2, \tag{23}$$

For the second term of Eq. 22, we can expand it by utilizing the $\ell_2$ norm characteristic of the Hessian matrix. It can be defined as follows:

$$\|H_f(x + \xi\epsilon\delta)(\epsilon\delta)^2\|_2 \leq \|\lambda_{Hmax}(\epsilon\delta)^2\|_2 \leq \lambda_{Hmax}\epsilon^2, \tag{24}$$

where $\lambda_{Hmax}$ is the maximum eigenvalue of the Hessian matrix. Substituting these bounds into the expression for the squared difference, we obtain:

$$\|f(x + \epsilon\delta) - f(x)\|_2^2 \leq \epsilon^2 \|J_f(x)\|_2^2 + \epsilon^3 \lambda_{Hmax} \|J_f(x)\|_2 + \frac{\epsilon^4 \lambda_{Hmax}^2}{4}, \tag{25}$$

Thus, the adversarial perturbation measure satisfies the following upper bound:

$$\mathcal{R}_{\text{adv}}(f, x, \epsilon) \leq \epsilon^2 \|J_f(x)\|_2^2 + O(\epsilon^2), \tag{26}$$

where $O(\epsilon^2)$ represents a higher-order infinitesimal of $\epsilon^2$. Since $\epsilon$ is a very small quantity, the term $O(\epsilon^2)$ in the formula can be neglected. Consequently, the theoretical upper bound of the adversarial perturbation is primarily dependent on the $\ell_2$ norm of $J_f(x)$. Specially, $J_f(x)$ can be expressed as the collection of gradients of each component of the function:

$$J_f(x) = \begin{bmatrix} \nabla f_1(x)^T \\ \nabla f_2(x)^T \\ \vdots \\ \nabla f_m(x)^T \end{bmatrix}. \tag{27}$$

The $\ell_2$ norm of the Jacobian matrix is related to the gradients through the following expression:

$$\|J_f(x)\|_2^2 = \lambda_{Jmax}(J_f(x)^T J_f(x)) = \lambda_{Jmax}\left( \sum_{i=1}^{m} \nabla f_i(x) \nabla f_i(x)^T \right). \tag{28}$$

Where $\lambda_{Jmax}$ denotes the maximum eigenvalue of the matrix, which reflects the largest possible stretching effect of the Jacobian matrix, directly linking the gradient magnitudes to the overall sensitivity of the network's output.

## C  ACTIVATION-BASED UPPER BOUND OF ADVERSARIAL ATTACK

**Theorem**: For a discrete spike pattern mapping $f : \mathbb{R}^n \to \mathbb{R}^m$, small perturbations $\varepsilon\delta$ around input $x$ induce a finite set of activation pattern transitions. The adversarial robustness upper bound can be approximated as:

$$R_{\text{adv}}(f, x, \varepsilon) \leq \varepsilon^2 \max_{1 \leq k \leq K} \|A_{\mathcal{A}_k}\|_{p \to 2}^2,$$

where $K$ denotes the number of activation regions intersecting the perturbation ball $B_\varepsilon(x)$, and $A_{\mathcal{A}_k} \in \mathbb{R}^{m \times n}$ is the affine transformation matrix for activation pattern $\mathcal{A}_k = \{(l, i) : u_i(x) \geq \theta_i\}$.

**Proof:** Consider an SNN represented by the function $f : \mathbb{R}^n \to \mathbb{R}^m$, where $\mathbb{R}^n$ denotes the input space and $\mathbb{R}^m$ denotes the output space. The adversarial perturbation measure at an input point $x \in \mathbb{R}^n$ with a perturbation radius $\varepsilon > 0$ is defined as:

$$R_{\text{adv}}(f, x, \varepsilon) = \max_{\|\delta\|_p \leq 1} \|f(x + \varepsilon\delta) - f(x)\|_2^2, \tag{29}$$

where $\delta$ represents the perturbation direction vector constrained by the $p$-norm unit ball. Given the piecewise-linear nature of the SNN, small perturbations $\varepsilon\delta$ around an input $x$ lead to a finite set of possible activation pattern changes. Specifically, the activation pattern at any input $x$ can be defined as:

$$A(x) = \{(l, i) : u_i^{(l)}(x) \geq \theta_i^{(l)}\}, \tag{30}$$

where $u_i^{(l)}(x)$ denotes the membrane potential of neuron $i$ at layer $l$, and $\theta_i^{(l)}$ denotes the corresponding firing threshold. Consequently, each distinct activation pattern $\mathcal{A}$ corresponds uniquely to a convex polyhedral region in the input space defined as:

$$R_{\mathcal{A}} = \{x \in \mathbb{R}^n : A(x) = \mathcal{A}\}. \tag{31}$$

Within any such region $R_{\mathcal{A}}$, the network behaves as an affine transformation described by:

$$f(x) = A_{\mathcal{A}} x + b_{\mathcal{A}}, \tag{32}$$

where $A_{\mathcal{A}} \in \mathbb{R}^{m \times n}$ and $b_{\mathcal{A}} \in \mathbb{R}^m$ are determined entirely by the network's weights and biases under the specific activation pattern $\mathcal{A}$. For sufficiently small perturbations $\delta$ ensuring $x + \varepsilon\delta \in R_{\mathcal{A}}$, the network's output change can explicitly be expressed as:

$$f(x + \varepsilon\delta) - f(x) = \varepsilon A_{\mathcal{A}}\delta. \tag{33}$$

Utilizing the properties of operator norms, we bound the change in network output within the given region by:

$$G(x) = \|f(x + \varepsilon\delta) - f(x)\|_2^2, \quad G(x) \leq \varepsilon^2 \|A_{\mathcal{A}}\|_{p\to2}^2, \tag{34}$$

where $\|A_{\mathcal{A}}\|_{p\to2}$ denotes the operator norm of the matrix $A_{\mathcal{A}}$ defined with respect to the input $p$-norm and output 2-norm. Considering all possible regions intersecting the perturbation ball $B_\varepsilon(x) = \{x + \varepsilon\delta : \|\delta\|_p \leq 1\}$, we derive the global bound for the adversarial perturbation measure as:

$$R_{\text{adv}}(f, x, \varepsilon) \leq \varepsilon^2 \max_{1 \leq k \leq K} \|A_{\mathcal{A}_k}\|_{p\to2}^2, \tag{35}$$

where $K$ represents the finite number of distinct activation regions intersecting $B_\varepsilon(x)$. Further analyzing the structure of the matrices $A_{\mathcal{A}_k}$, one observes that the sensitivity of these matrices primarily depends on neurons whose membrane potentials $u_i^{(l)}(x)$ are near their firing thresholds $\theta_i^{(l)}$. Consequently, larger distances between membrane potentials and thresholds indicate greater activation stability, leading to smaller variations in the affine transformation $A_{\mathcal{A}_k}$ and ultimately reducing the operator norm $\|A_{\mathcal{A}_k}\|_{p\to2}$. Formally stated, as the absolute difference between the membrane potential $u_i^{(l)}(x)$ and the threshold $\theta_i^{(l)}$ increases, the adversarial perturbation measure strictly decreases:

$$|u_i^{(l)}(x) - \theta_i^{(l)}| \uparrow \implies R_{\text{adv}}(f, x, \varepsilon) \downarrow . \tag{36}$$

## D  PROOF OF THE PROBABILITY FOR SPIKING NEURONS' STATE FLIPPING

**Theorem**: Consider a spiking neuron with membrane potential $V[t]$ at time $t$, firing threshold $V_{\text{th}}$, and subject to Gaussian white noise perturbation $\eta[t] \sim \mathcal{N}(0, \sigma^2)$. The probability $P_{\text{flip}}$ of the neuron's state transition (flipping) is given by:

$$P_{\text{flip}} = \begin{cases} \Phi\left(\frac{V_{\text{th}} - V[t]}{\sigma}\right), & \text{if } V[t] \geq V_{\text{th}}, \\ 1 - \Phi\left(\frac{V_{\text{th}} - V[t]}{\sigma}\right), & \text{if } V[t] < V_{\text{th}}, \end{cases}$$

where $\Phi(\cdot)$ denotes the cumulative distribution function (CDF) of the standard normal distribution.

**Proof**: Let us consider the stochastic dynamics of the neuron's membrane potential under noise perturbation. Define the noise-perturbed membrane potential as $H' = V[t] + \eta[t]$, where $\eta[t] \sim \mathcal{N}(0, \sigma^2)$ represents additive Gaussian white noise. The neuron's state transition probability depends on whether this perturbed potential crosses the threshold $V_{\text{th}}$. We analyze this probability by considering two distinct cases.

For Case 1, when $V[t] \geq V_{\text{th}}$, the deterministic dynamics would result in the neuron firing (output state = 1). However, the presence of noise can induce a transition to the non-firing state (0). This flip occurs if and only if the perturbed potential falls below threshold:

$$H' < V_{\text{th}} \iff V[t] + \eta[t] < V_{\text{th}} \iff \eta[t] < V_{\text{th}} - V[t]. \tag{37}$$

Since $\eta[t]$ follows a normal distribution with mean 0 and variance $\sigma^2$, we can standardize this inequality. The probability of transition from state 1 to 0 is:

$$P_{1\to0} = P(\eta[t] < V_{\text{th}} - V[t]) = \Phi\left(\frac{V_{\text{th}} - V[t]}{\sigma}\right), \tag{38}$$

where the last equality follows from the definition of the standard normal CDF.

For Case 2, when $V[t] < V_{\text{th}}$, the deterministic dynamics would result in no firing (output state = 0). A transition to the firing state (1) occurs when noise pushes the membrane potential above threshold:

$$H' \geq V_{\text{th}} \iff V[t] + \eta[t] \geq V_{\text{th}} \iff \eta[t] \geq V_{\text{th}} - V[t]. \tag{39}$$

Following the same probabilistic reasoning, and noting that $P(\eta[t] \geq x) = 1 - P(\eta[t] < x)$ for any $x$, the probability of transition from state 0 to 1 is:

$$P_{0 \to 1} = P(\eta[t] \geq V_{\text{th}} - V[t]) = 1 - \Phi\left(\frac{V_{\text{th}} - V[t]}{\sigma}\right). \tag{40}$$

Combining these cases yields the desired expression for $P_{\text{flip}}$. Note that this result naturally captures the intuition that the probability of state transition decreases as the membrane potential moves further from the threshold in either direction, due to the monotonicity properties of $\Phi$.

## E  PROOF OF THE PROBABILITY FOR NOISY-LIF'S STATE FLIPPING

**Theorm:** Consider a Noisy-LIF neuron with membrane potential $V[t]$, Gaussian noise $\xi[t] \sim \mathcal{N}(0, \sigma^2)$, and threshold $V_{\text{th}}$. The probability of firing can be expressed as:

$$P(S_l = 1 \mid V[t], \xi[t]) = 1 - \Phi\left(\frac{V_{\text{th}} - (V[t] + \xi[t])}{\sigma}\right),$$

where $\Phi(\cdot)$ is the CDF of the standard normal distribution. As the $\sigma$ increases, the probability density function $\phi$ becomes broader, reducing the sensitivity to small variations in $V[t]$.

**Proof:** The firing condition in the Noisy-LIF neuron model is given by the inequality $V[t] + \xi[t] \geq V_{\text{th}}$. To analyze the firing probability conditioned on the membrane potential $V[t]$ and the noise $\xi[t]$, we start by rewriting the firing condition in terms of the standard normal distribution:

$$P(S_l = 1 \mid V[t], \xi[t]) = P(\xi[t] \geq V_{\text{th}} - V[t]) = 1 - \Phi\left(\frac{V_{\text{th}} - V[t]}{\sigma}\right). \tag{41}$$

Differentiating this probability with respect to $V[t]$ gives us the sensitivity of the firing probability to changes in the membrane potential:

$$\frac{\partial P(S_l = 1 \mid V[t], \xi[t])}{\partial V[t]} = \frac{1}{\sigma}\phi\left(\frac{V_{\text{th}} - V[t]}{\sigma}\right), \tag{42}$$

where $\phi$ is the probability density function of the standard normal distribution. This derivative indicates how a small change in $V[t]$ affects the probability of firing. To understand the impact of $\sigma$ on the sensitivity, consider that as $\sigma$ increases, the width of $\phi$ also increases, making it flatter. This change results in a decrease in the magnitude of the derivative, indicating reduced sensitivity to small changes in $V[t]$.

In this research, the training process for all experiments extends over a duration of 300 epochs. To address potential vanishing or exploding gradients, batch normalization techniques are integrated throughout the network architecture. The LIF neurons in the experiments are configured with a decay factor of 0.5 and a threshold of 1. The proposed noisy LIF neurons incorporate Gaussian noise with a mean of 0 and a variance of 0.4. All experiments were conducted with 300 training iterations, repeated thrice, and the reported results are the averages of these three runs. For optimization, we implement the stochastic gradient descent (SGD) algorithm, starting with a learning rate set at 0.1. The adjustment of the learning rate follows a cosine annealing strategy. All experiments are performed on a PyTorch framework, facilitated by the computational power of an NVIDIA RTX 4090 GPU.

## F  VISUALIZATION OF GRADIENT SPARSITY AND LOSS LANDSCAPE

To assess the effectiveness of TGO in adversarial environments, we visualized the Gradient Sparsity and loss landscape of the BPTT-based WR16 model under RFSGM attack, comparing TGO optimization with the vanilla SNN. The Gradient Sparsity quantifies the gradient $\nabla_x f_y$ between the label and the input image.

**Gradient Sparsity**: The gradient visualization begins with an input image $\mathbf{I}_i \in \mathbb{R}^{C \times H \times W}$, where $C$ denotes the number of channels (e.g., RGB), and $H \times W$ represents the image's spatial resolution.

A pre-trained model $f(\mathbf{I}; \theta)$, parameterized by $\theta$, maps the input image to a probability distribution over $K$ classes, denoted as $\mathbf{p} \in \mathbb{R}^K$. The prediction process is given by:

$$\mathbf{p} = f(\mathbf{I}_i; \theta), \quad \mathbf{p}[y_i] = \frac{\exp(z_{y_i})}{\sum_{k=1}^{K} \exp(z_k)}, \tag{43}$$

where $z_k$ is the pre-activation value for class $k$ prior to the application of the softmax function. The model is optimized using the cross-entropy loss between the predicted probability distribution and the true label $y_i$, formulated as:

$$\mathcal{L}(\mathbf{I}_i, y_i) = -\log \mathbf{p}[y_i] = -\log \left( \frac{\exp(z_{y_i})}{\sum_{k=1}^{K} \exp(z_k)} \right). \tag{44}$$

To assess the sensitivity of the model's predictions to the input, the gradient of the loss with respect to the input image is computed, resulting in a gradient tensor $\mathbf{g}_i \in \mathbb{R}^{C \times H \times W}$, expressed as:

$$\mathbf{g}_i = \frac{\partial \mathcal{L}(\mathbf{I}_i, y_i)}{\partial \mathbf{I}_i}, \quad \mathbf{g}_i[c, x, y] = \frac{\partial}{\partial \mathbf{I}_i[c, x, y]} \left( -\log \mathbf{p}[y_i] \right), \tag{45}$$

where $c$, $x$, and $y$ correspond to the channel and spatial coordinates of the image. We compute the gradient $\nabla_x f_y(x)$ under two scenarios: (1) a vanilla SNN, and (2) a TGO-optimized SNN, both trained using BPTT. As shown in Fig.5, we visualize $\nabla_x f_y(x)$ for both models on CIRAF00. The experimental results clearly indicate that, compared to the vanilla SNN, the TGO-optimized SNN exhibits sparser gradients with respect to the input image, demonstrating the effectiveness of membrane potential constraints in increasing gradient sparsity in SNNs. These findings also validate that sparse gradients contribute to enhancing the robustness of SNNs.

**Loss Landscape**: The process of visualizing the loss landscape starts with a pre-trained model parameterized by $\theta \in \mathbb{R}^M$, where $M$ represents the total number of parameters in the model. The performance of the model is quantified using a loss function $\mathcal{L}(\theta)$, defined over a dataset of $N$ samples as:

$$\mathcal{L}(\theta) = \frac{1}{N} \sum_{i=1}^{N} \ell(f(\mathbf{I}_i; \theta), y_i), \quad \ell(f(\mathbf{I}_i; \theta), y_i) = -\log \mathbf{p}[y_i], \tag{46}$$

where $f(\mathbf{I}_i; \theta)$ denotes the output of the model for input $\mathbf{I}_i$, $\ell$ is the sample-wise loss (e.g., cross-entropy), and $(\mathbf{I}_i, y_i)$ are the input and label for the $i$-th sample. The predicted probability distribution $\mathbf{p} \in \mathbb{R}^K$ is computed via softmax as:

$$\mathbf{p}[y_i] = \frac{\exp(z_{y_i})}{\sum_{k=1}^{K} \exp(z_k)}, \quad z_k = f_k(\mathbf{I}_i; \theta), \tag{47}$$

where $z_k$ is the pre-activation value for class $k$, and $K$ is the total number of classes. To visualize the local geometry of $\mathcal{L}(\theta)$ around a specific parameter set $\theta$, two directions $\mathbf{d}_1, \mathbf{d}_2 \in \mathbb{R}^M$ are defined, satisfying:

$$\|\mathbf{d}_1\| = \|\mathbf{d}_2\| = 1, \quad \mathbf{d}_1^\top \mathbf{d}_2 = 0. \tag{48}$$

Typically, $\mathbf{d}_1$ is chosen as the gradient direction:

$$\mathbf{d}_1 = \frac{\nabla_\theta \mathcal{L}(\theta)}{\|\nabla_\theta \mathcal{L}(\theta)\|}, \quad \nabla_\theta \mathcal{L}(\theta) = \frac{\partial \mathcal{L}(\theta)}{\partial \theta}, \tag{49}$$

while $\mathbf{d}_2$ is sampled randomly and orthogonalized to $\mathbf{d}_1$. The perturbed parameter vector is then expressed as:

$$\theta' = \theta + \alpha \mathbf{d}_1 + \beta \mathbf{d}_2, \tag{50}$$

where $\alpha, \beta \in \mathbb{R}$ control the magnitudes of the perturbations along $\mathbf{d}_1$ and $\mathbf{d}_2$. At each perturbation point $(\alpha, \beta)$, the loss is computed as:

$$\mathcal{L}(\theta') = \frac{1}{N} \sum_{i=1}^{N} \ell(f(\mathbf{I}_i; \theta + \alpha \mathbf{d}_1 + \beta \mathbf{d}_2), y_i). \tag{51}$$

The parameter space is discretized into a grid of perturbation values $\alpha \in [-\alpha_{\max}, \alpha_{\max}]$ and $\beta \in [-\beta_{\max}, \beta_{\max}]$, such that the loss values form a matrix:

$$\mathbf{L} = [\mathcal{L}(\theta + \alpha \mathbf{d}_1 + \beta \mathbf{d}_2)] \in \mathbb{R}^{n_\alpha \times n_\beta}, \tag{52}$$

where $n_\alpha$ and $n_\beta$ are the number of grid points along the $\alpha$ and $\beta$ directions, respectively. Finally, $\mathbf{L}$ is visualized as either a 3D surface or a 2D contour plot:

$$\mathcal{L}(\alpha, \beta) = \mathcal{L}(\theta + \alpha\mathbf{d}_1 + \beta\mathbf{d}_2) \in \mathbb{R}^3, \tag{53}$$

where the gradient of the loss surface can reveal the flatness, sharpness, or other geometric properties of $\mathcal{L}(\theta)$ in the vicinity of the parameter set. As shown in the left part of Fig. **??**, the 3D loss landscape reveals that the TGO-optimized model maintains stable loss convergence, demonstrating robust performance. In contrast, the vanilla SNN exhibits localized reverse peaks in the loss landscape under attack. This instability is due to the FGSM gradient-based attack, which perturbs the input along the loss gradient, causing significant fluctuations in the loss. Specifically, FGSM computes the gradient of the loss with respect to the input image and perturbs the input to maximize the loss, pushing the vanilla SNN into regions of the loss landscape that are highly sensitive to small perturbations. In contrast, the TGO-optimized model exhibits smoother transitions in the loss landscape, indicating that its optimization enhances stability against adversarial attacks.

## G    LIMITATIONS

The limitations of this study include the performance evaluation of Our TGO method on larger model architectures, primarily because existing research predominantly utilizes these specific network structures and their corresponding datasets. For comparative consistency, we maintained these established architectures. Additionally, we have not addressed deployment challenges related to hardware transitions, nor conducted robustness testing in authentic edge-computing adversarial environments. These limitations will be addressed in future research. The experimental results presented in this paper are reproducible, with detailed explanations of model training and configuration provided in the main text and supplemented in the appendix. Our code and models will be made publicly available on GitHub upon acceptance of this paper.

## H    USE OF LARGE LANGUAGE MODEL

In preparing this manuscript, we utilize a large language model (LLM) solely to aid and polish the writing. The LLM is used for grammar checking, language refinement, and improving clarity of expression. It does not contribute to the formulation of research ideas, methodology, experiments, data analysis, or conclusions. All presented in this paper is entirely the work of the authors.

