# OpenReview forum: "Robust Spiking Neural Networks Against Adversarial Attacks"
_ICLR.cc/2026/Conference — ICLR 2026 Poster_

### Official Review · Reviewer_23Ay · 2025-10-20

**Soundness:** 1
**Presentation:** 2
**Contribution:** 1
**Rating:** 2
**Confidence:** 5

**Summary:**

This paper introduces a Threshold Guarding Optimization (TGO) approach against the adversarial robustness problem of directly-trained spiking neural network (SNNs) architectures. The method relies on defending neurons that have membrane potentials very close to the firing threshold, which SNN adversaries essentially exploit. The defense mechanism uses (1) layerwise loss regularizers that move neuron membrane potentials away from the firing thresholds, thus effectively creates sparsity in the surrogate gradients, and (2) noisy LIF neurons to reduce the likelihood of state-flipping under minimal adversarial noise disturbances. Experiments on CIFAR-10/100 with VGG-11 and WRN-16 architectures demonstrate that TGO is effective when combined with adversarial training based methods.

**Strengths:**

- The paper analyzes a clear cause of adversarial robustness of SNNs, i.e., neurons having threshold-neighboring membrane potentials for test samples. This is also the case for ANN neuron activations, which naturally aligns well in this paper.

- The narrative and descriptions of the TGO methodology is clear.

**Weaknesses:**

- Experimental evaluations are rather ambigious to draw any conclusions (e.g., noisy inference processes, potentially missing random restarts and EoT, weak attack strengths). It also appears like with simple BPTT, TGO is not highly effective as a standalone defense. Furthermore, surrogate gradient ensemble evaluations are missing, which should have been the rigorous attack baseline for SNN adversarial robustness.

- There are several missing details in terms of hyperparameters, evaluation settings and consistency of the results from the main text to the appendix (where there are really informative results existing).

**Questions:**

- One of the most critical components of the defense is the use of randomness and injecting noise during inference. This is well-known to significantly prohibit accurate adversarial robustness evaluations. However, authors state that they only employ EoT to investigate this in Appendix B, although EoT and reliably evaluated robust accuracies under random restarts should have been present in all evaluations of the main manuscript.

- The paper talks about directly-trained SNNs and surrogate gradient choices, but never really states the exact surrogate gradient function used in training and evaluation of their models?

- Following the above question, there is also already an established surrogate gradient ensemble based SNN attack, which the paper does not consider: https://openreview.net/pdf?id=I8FMYa2BdP . It essentially aims to evaluate directly trained SNNs more reliably, by allowing the white-box adversaries to try out different surrogate gradient functions for more stable attacks. This aligns with basic security principles, where white-box adversaries should have complete access and capabilities in evaluating models. I would expect the authors to evaluate their "adversarially robust SNNs" under such ensemble attacks, by reporting robust accuracies under surrogate gradient adaptive adversaries.

- The notion of imposing gradient sparsity was already the main idea in the SR approach. How is the present paper different?

- Why are the naive attacks in Table 1 for CIFAR100/WRN-16 with AT+TGO(Ours) more effective than the results in Table 5 when EoT is added? The main idea in EoT is to obtain more rigorous adversaries, without making the attack weaker?

- In Table 2 APGD_CE row, increasing the steps from 80 to 100 makes the attacks slightly weaker, given the numbers in the table. This should not happen in any case. There is some ambiguity regarding the convergence of attacks. How is the attack success calculated with increasing number of iterations?

- Are there any other datasets/architectures that this approach would scale and advance the SoTA? Can we use this method besides VGG and WRN type of spiking networks, or on images larger than 32x32 resolution?

- In general, evaluations are also demonstrated at a stronger perturbation radius than in the training phase. Also, is there the usual "random restarts" idea implemented in these multi-step PGD attacks?

- No hyperparameter details are present. What is the lambda hyperparameter value in the main results? It is not clearly described anywhere, and none of the results for lambdas match consistently between Table 1 and Table 7 either, there appears to be some rows shifted or something. Overall, it is not possible to extract accurate information from the current presentation of results either.

- The results on DVS datasets are only present in Appendix A, very briefly without details. Can the authors elaborate further, how they actually implemented this? These attacks should be fundamentally different to implement, since the inputs are binary. How does the perturbation strength work here for instance? Also, there is a typo there: DVS-CRIAF10 -> DVS-CIFAR10.

- Regarding Figure 4 right side loss landscapes - It is not described how two-dimensional loss landscape visualizations are generated?

---

> ### Author Response · Authors · 2025-11-22
>
> We sincerely appreciate your recognition of the innovative aspects of our proposed TGO method. Below, we provide detailed responses addressing each of your **concerns**.
>
> ## Weakness 1: One of the most critical components of the defense is the use of randomness and injecting noise during inference. This is well-known to significantly prohibit accurate adversarial robustness evaluations. However, authors state that they only employ EoT to investigate this in Appendix B, although EoT and reliably evaluated robust accuracies under random restarts should have been present in all evaluations of the main manuscript.
>
> Thank you for your valuable feedback. We employ standard attacks in the main experiments to ensure fair comparison with existing works (e.g., SToG[1]), which also exhibit inherent randomness(Bernoulli distribution-based stochastic gating). This consistent protocol ensures valid and comparable evaluations across methods. Moreover, We acknowledge that the randomness introduced by Noisy-LIF may affect robustness assessment. Therefore, we conducted supplementary experiments using EOT (Expectation Over Transformation) attacks specifically on our proposed TGO method. In this experiment, the hyperparameter $\lambda$ is set to 0.6.
>
> |model|PGD10|PGD20|PGD30|PGD40|PGD50|
> |:-:|:-:|:-:|:-:|:-:|:-:|
> |AT|11.12|9.22|7.50|7.06|6.72|6.82|
> |AT+TGO(Standed)|29.48|27.72|26.21|25.89|24.74|
> |TGO+AT(EoT)|26.59|24.57|22.68|22.43|22.28|
>
> The results show that while adversarial robustness decreases under EOT compared to standard attack.  EoT optimizes a single perturbation $\delta$ to be effective over an entire family of transformed inputs, rather than a single fixed sample. This defines a strictly stronger attack constraint, leading to more damaging perturbations and consequently lower model performance. But TGO method still achieve significant performance gains over baseline AT, even under this stricter evaluation.
>
>
>
>
> ## Weakness 2: The paper talks about directly-trained SNNs and surrogate gradient choices, but never really states the exact surrogate gradient function used in training and evaluation of their models?
>
> We thank the reviewer for pointing out this oversight. All experiments in this paper employ the rectangular surrogate gradient function during training. In response to the reviewer's concern regarding different surrogate gradient functions, we conducted additional comparative experiments. As shown in the Table, we evaluated various surrogate gradient functions using WideResNet-16 on CIFAR-100. In this experiment, the hyperparameter $\lambda$ is set to 0.4.
>
> | model    | Surrogate Gradient | Clean | FGSM | RFGSM | PGD7 | PGD10 | PGD20 | PGD40 |
> |----------|--------------------|-------|------|-------|------|-------|-------|-------|
> | TGO+AT   | Tria               | 64.49 | 41.99| 55.01 | 27.78| 22.75 | 15.99 | 14.59 |
> | TGO+AT   | Atan               | 65.25 | 43.52| 56.38 | 29.15| 24.21 | 17.36 | 15.83 |
> | TGO+AT   | Rect               | 64.86 | 42.68| 55.74 | 28.51| 23.54 | 16.73 | 15.25 |
>
>
>  The results indicate that the choice of surrogate gradient function has minimal impact on TGO performance. Both Atan and Rect achieve slightly higher accuracy in the clean setting, but their performance gap under adversarial attacks is minimal.
>
>
>
>
>
>
>
>
>
> ## Weakness 3: Following the above question, there is also already an established surrogate gradient ensemble based SNN attack, which the paper does not consider: https://openreview.net/pdf?id=I8FMYa2BdP .
>
>
> We thank the reviewer for suggesting the surrogate gradient ensemble-based SNN attack method[2]. This approach allows white-box attackers to explore diverse surrogate gradient functions, adhering to the principle that white-box attacks require full model access for reliable robustness evaluation.
>
> | Method | Atan | Rect | Tria |
> |--------|-----|------|------|
> | AT | 18.58  | 19.39 | 11.14 |
> | AT+TGO | 35.52| 38.23 | 41.08 |
>
> We have conducted experiments using this ensemble attack to evaluate TGO. Results show that while the attack impacts TGO's performance, the core defense mechanisms remain effective. This resilience stems from two factors: (1) TGO's optimization targets the inherent vulnerability of neurons near the firing threshold—a generalized SNN weakness independent of specific surrogate gradient types; (2) its probabilistic firing mechanism and membrane potential constraints effectively mitigate perturbations from multiple gradient variants. We will include these experimental results and analyses in the revised manuscript to further validate TGO's robustness under complex adversarial scenarios. We appreciate the reviewer's valuable feedback.

---

> ### Author Response · Authors · 2025-11-22
>
> ## Weakness 4: The notion of imposing gradient sparsity was already the main idea in the SR approach. How is the present paper different?
>
> Thank you for your valuable feedback. We clarify that the "activation sparsification" mentioned in [2] is not our method's core objective. TGO's fundamental principle is to reduce the proportion of spike state alterations induced by adversarial noise, with activation sparsification being merely a potential manifestation rather than the primary goal. Specifically, TGO does not aim to reduce neuronal firing rates; instead, it employs membrane potential constraints to drive potentials away from the firing threshold (either substantially above or below). This mechanism fundamentally differs from SR methods. To highlight this distinction, we conducted the following ablation study:
>
> |model|FGSM|PGD10|PGD20|PGD30|PGD40|PGD50|PGD80|
> |:-:|:-:|:-:|:-:|:-:|:-:|:-:|:-:|
> |AT|21.18|9.22|7.50|7.06|6.72|6.82|6.67|
> |SR[3]+Noisy-LIF+AT|32.47|21.66|19.98|19.14|18.78|18.79|18.86|
> |TGO+AT|42.86|26.59|24.57|22.68|22.43|22.28|21.96|
>
> As shown in the above Table, we validated the feasibility of combining Noisy-LIF with SR methods in early investigations. While technically viable, performance improvements were limited. Experimental results demonstrate that TGO significantly outperforms the latter approach. This is primarily attributed to the mechanistic synergy between TC and Noisy-LIF—both fundamentally aim to reduce the proportion of spike state alterations induced by adversarial noise.
>
>
> ## Weakness 5: Why are the naive attacks in Table 1 for CIFAR100/WRN-16 with AT+TGO(Ours) more effective than the results in Table 5 when EoT is added? The main idea in EoT is to obtain more rigorous adversaries, without making the attack weaker?
>
> We apologize for the misunderstanding, which resulted from an oversight on our part. In our TGO method, there is a hyperparameter $\lambda$ that controls the strength of the membrane potential constraint. As shown bolew:
> $$
>     \mathcal{L}(\mathbf{x}, \lambda) = \mathcal{L}oss(\mathbf{x}) + \lambda \sum_l \mathcal{C}(V(t)_l).
> $$
> A larger $\lambda$ increases adversarial robustness but reduces clean accuracy, making the choice of $\lambda$ a trade-off. we performed experiments on CIFAR-100 using the WRN-16 architecture. In Table 1, the TGO method uses $\lambda = 0.4$, which leads to higher clean accuracy (60.37%) compared to SR+AT. However, in Table 5, where we test different attack radii, $\lambda = 0.6$ is used, resulting in lower clean accuracy than TGO+AT. We will clarify the choice of $\lambda$ in the revised version. Thank you again for pointing out this oversight.
>
>
>
> ## Weakness 6: In Table 2 APGD_CE row, increasing the steps from 80 to 100 makes the attacks slightly weaker, given the numbers in the table. This should not happen in any case. There is some ambiguity regarding the convergence of attacks. How is the attack success calculated with increasing number of iterations?
> Thank you for noting this observation. The slightly higher APGD100 compared to APGD80 can be attributed to two factors:
> First, as you noted, TGO introduces stochasticity at the membrane potential level. Since this experiment does not employ EOT (Expectation Over Transformation) constraints, APGD100 exhibits greater susceptibility to random fluctuations.
> Second, the APGD-CE attack converges at approximately 50 iterations, with subsequent APGD80 and APGD100 metrics remaining nearly stable relative to APGD50. In this converged state, the model demonstrates increased sensitivity to stochastic perturbations, resulting in minor performance variations.
> Therefore, the slight elevation of APGD100 over APGD80 represents a normal consequence of post-convergence stochasticity rather than a systematic deficiency of the method.

---

> ### Author Response · Authors · 2025-11-22
>
> ## Weakness 7: Are there any other datasets/architectures that this approach would scale and advance the SoTA? Can we use this method besides VGG and WRN type of spiking networks, or on images larger than 32x32 resolution?
>
> We present analysis focuses exclusively on the performance comparison between baseline and baseline+TGO configurations following fine-tuning under RFGSM adversarial attacks on Imagenet-1K with resnet34: :https://github.com/Nikb033/adversarial-attacks-resnet34-imagenet. as demonstrated in the following table:
> |model| Noise Level (ε) | Top-1 Error (%) | Top-5 Error (%) |
> |-----------------|-----------------|-------------|-------------|
> |Base|  0.01            | 89.75                         | 65.63     |
> |Base| 0.03            | 96.16                         | 73.84     |
> |Base+TGO(Ours)| 0.01            | 77.18       | 46.42       |
> |Base+TGO(Ours)| 0.03            | 85.67      | 60.76       |
>
> As shown in the above Table, TGO effectively improves SNN adversarial robustness on ImageNet-1K compared to based model.
>
> ## Weakness 8: In general, evaluations are also demonstrated at a stronger perturbation radius than in the training phase. Also, is there the usual "random restarts" idea implemented in these multi-step PGD attacks?
> We trained the model with a perturbation radius of 2 and evaluated it under perturbation radii of 2, 4, 6, and 8, as shown in Fig.3 in our paper.
>
> Additionally, we replicate the Multi-Targeted PGD (MTPGD) method from [4] with the following parameter settings: epsilon=8/255, alpha=2/255, including **random initialization and clipping**. The comparison results are shown below:
>
> | model | MTPGD10 | MTPGD20 | MTPGD30 | MTPGD40 | MTPGD50 | MTPGD80 | MTPGD100 |
> |:------|:-------:|:-------:|:-------:|:-------:|:-------:|:-------:|:--------:|
> | AT | 11.61 | 9.59 | 9.27 | 8.97 | 8.57 | 8.49 | 8.49 |
> | SR[3]+AT | 23.17 | 21.79 | 21.54 | 21.39 | 21.45 | 21.28 | 20.81 |
> | TGO+AT(Ours) | 43.49 | 36.88 | 34.04 | 31.45 | 30.70 | 30.10 | 29.76 |
>
> The results show that our method significantly outperforms pure AT and SR method against APGD and MTPGD attacks. Although it may not yet match the latest ANN methods, it represents a meaningful effort to explore the robustness of spiking neurons.
>
> ## Weakness 9: No hyperparameter details are present. What is the lambda hyperparameter value in the main results? It is not clearly described anywhere, and none of the results for lambdas match consistently between Table 1 and Table 7 either, there appears to be some rows shifted or something. Overall, it is not possible to extract accurate information from the current presentation of results either.
> As with Question 5, we did not explicitly indicate λ in the table. A larger $\lambda$ increases adversarial robustness but reduces clean accuracy, making the choice of $\lambda$ a trade-off. we performed experiments on CIFAR-100 using the WRN-16 architecture. The results are shown below.
> |model|$\lambda_{max}$|Clean|FGSM|RFGSM|PGD7|PGD10|PGD20|PGD30|PGD40|PGD50|
> |:-:|:-:|:-:|:-:|:-:|:-:|:-:|:-:|:-:|:-:|:-:|
> |TGO+AT| 0.2  | 66.93 | 34.53 | 53.34 | 18.93 | 14.32 | 9.71 | 8.00 | 7.50 | 7.14 |
> |TGO+AT| 0.3  | 65.80 | 39.02 | 54.32 | 24.23 | 19.35 | 13.58 | 11.92 | 10.74 | 10.50 |
> |TGO+AT| 0.4  | 64.49| 41.99 | 55.01 | 27.78 | 22.75 | 15.99 | 14.59 | 14.12 | 13.79 |
> |TGO+AT| 0.6  |59.56| 48.27 | 54.37 | 41.08 | 36.65 | 31.15 | 29.19 | 28.46 | 28.12 |
>
> In Table 1, the TGO method uses $\lambda = 0.4$, which leads to higher clean accuracy (60.37%) compared to SR+AT. However, in Table 7, where we test different attack radii, $\lambda = 0.6$ is used, resulting in lower clean accuracy than TGO+AT. We will clarify the choice of $\lambda$ in the revised version. Thank you again for pointing out this oversight.

---

> ### Author Response · Authors · 2025-11-22
>
> ## Weakness 10: The results on DVS datasets are only present in Appendix A, very briefly without details. Can the authors elaborate further, how they actually implemented this? These attacks should be fundamentally different to implement, since the inputs are binary. How does the perturbation strength work here for instance? Also, there is a typo there: DVS-CRIAF10 -> DVS-CIFAR10.
> In the appendix, we briefly describe the application of our method against attacks specifically designed for DVS data, which includes two scenarios.
> Scenario 1: The DVS data is first compressed into frames and then subjected to conventional adversarial attacks, similar to those used for static images（CIFAR-100. For static images in SNNs, the same image is repeatedly input at each timestep, whereas for DVS data, compressed event frames are input at each timestep. Despite this difference in input representation, the attack settings and procedures remain consistent with those used for attacking static images in SNNs.
>
> Scenario 2: We evaluate the performance of our TGO method under a state-of-the-art attack specifically designed for event-stream data. Specifically, we applied the event-based attack [5] to evaluate WideResNet-16 on DVS-CIFAR10. The core idea of this attack through three key strategies: (1) discrete-to-continuous relaxation via probabilistic sampling, (2) selective perturbation on critical event positions, and (3) sparsity-preserving regularization. The attack success rate (ASR) for AT is 44.32%, while AT+TGO achieves 31.0%. A lower ASR indicates higher robustness, demonstrating the effectiveness of the TGO method against event-based attacks.
>
> ## Weakness 11: Regarding Figure 4 right side loss landscapes - It is not described how two-dimensional loss landscape visualizations are generated?
>
> Figure 4 compares the loss landscapes of the TGO-optimized model and the baseline model under FGSM attack for correctly classified samples. Notably, the TGO-optimized model preserves a distinct local minimum, while the baseline model exhibits noisy and irregular contour lines. **The 2D loss landscape visualization is constructed as follows:** Given a trained model with parameters $\theta_0$, we randomly generate two direction vectors $\delta_1$ and $\delta_2$ (typically normalized). We then sample points on the 2D plane spanned by these directions:
>
> $$\theta(\alpha, \beta) = \theta_0 + \alpha \delta_1 + \beta \delta_2$$
>
> where $\alpha, \beta \in [-a, a]$ define the coordinate range. For each sampled point $(\alpha, \beta)$, we compute the loss of the corresponding parameters $\theta(\alpha, \beta)$ on the given data. The resulting contour plot, with $(\alpha, \beta)$ as coordinates and loss as height, visualizes the loss geometry in the local neighborhood of the parameter space.
>
>
>
> ## Reference：
>
>
> [1] Ding, Jianhao, et al. "Enhancing the robustness of spiking neural networks with stochastic gating mechanisms." Proceedings of the AAAI Conference on Artificial Intelligence. Vol. 38. No. 1. 2024.
>
> [2] Özdenizci O, Legenstein R. Adversarially robust spiking neural networks through conversion[J].
>
>
> [3] Enhancing adversarial robustness in SNNs with sparse gradients, ICML, 2024.
>
> [4] An alternative surrogate loss for pgd-based adversarial testing. 2019.
>
> [5] Exploring Vulnerabilities in Spiking Neural Networks: Direct Adversarial Attacks on Raw Event Data, ECCV, 2024.

---

### Official Review · Reviewer_FMzT · 2025-10-27

**Soundness:** 3
**Presentation:** 2
**Contribution:** 3
**Rating:** 4
**Confidence:** 3

**Summary:**

This paper proposes a training strategy, termed TGO, to enhance the robustness of spiking neural networks (SNNs) against adversarial attacks such as FGSM, PGD, and their variants. The main idea is to constrain the membrane potential to remain sufficiently distant from the firing threshold, thereby reducing sensitivity to perturbations. In addition, the strategy introduces neuron-level perturbations (NLIF) and regularizes the probability of spike flipping. Experiments on the CIFAR-100 dataset demonstrate the effectiveness of the proposed approach.

**Strengths:**

**S1.** The idea is straightforward and easy to understand.

**S2.** The authors conduct extensive experiments, including Expectation over Transformation (EoT) and loss landscape analysis.

**S3.** As shown in the tables, TGO achieves the best robustness performance compared with state-of-the-art (SOTA) training strategies.

**Weaknesses:**

**W1.** The explanation of the idea is unnecessarily complicated. In particular, Theorem 2 seems redundant — it is difficult to follow due to the heavy notation, and after reading the proof in the appendix, it appears to be a straightforward extension of Theorem 1.

**W2.** I believe the proof of Theorem 3 may be incorrect. According to Appendix E, the flipping probability from 1 to 0 should be expressed as the conditional probability and the same applies to the flipping probability from 0 to 1.
$$
P_{1\rightarrow 0} = P(\eta[t] < V_{th} - V[t] | V[t] \geq V_{th}).
$$

**W3.** Theorem 1 is proved under an $\ell_2$ constraint, whereas the experiments are conducted with $\ell_\infty$ perturbations. This inconsistency raises questions about the necessity and practical relevance of the theorem.

**W4.** According to Eq. (7), the constraint loss needs to be computed in a layer-wise manner. Meanwhile, the NLIF module introduces perturbations at every layer. How efficient is TGO in terms of computation compared with other training strategies? Does the training time increase significantly as the network depth grows?

**W5.** The paper does not specify the value of $\delta$ in Eq. (7) or explain how it was chosen. Please clarify it.

**W6.** The paper contains some typos, though they do not affect my overall rating. A few examples are listed below:

1. Line 182: $|J_f(x)|_2^2$.
2. In Theorem 1, $\eta$ has a mean of 0, whereas in Lines 316–317 and Figure 2(d), another mean $\mu$ appears.
3. In Eq. (11), it seems that $P$ should be used instead of $\Delta P$.
4. The clean accuracy is not highlighted in Table 1.
5. In Appendix C, the proof is written under the condition $||\delta||_2 \leq 1$ rather than $||\delta||_p \leq 1$.

**Questions:**

Please address W2–W5 in the Weakness section. I will consider increasing my score if these concerns are fully addressed.

---

> ### Author Response · Authors · 2025-11-21
>
> We sincerely appreciate your recognition of the innovative aspects of our proposed TGO method. Below, we provide detailed responses addressing each of your **concerns**.
>
> ## Weakness 1: The explanation of the idea is unnecessarily complicated. In particular, Theorem 2 seems redundant — it is difficult to follow due to the heavy notation, and after reading the proof in the appendix, it appears to be a straightforward extension of Theorem 1.
> **A**:Thank you for your comment. Theorem 2 complements Theorem 1 by addressing its limitations. While Theorem 1 expands $f(x)$ using a Taylor series, the discrete nature of SNNs makes $f(x)$ non-differentiable. Even with surrogate gradients like sigmoid or arctan, the output remains continuous, leading to mismatches with actual layer outputs. Therefore, Theorem 1 alone is insufficient.
>
> Theorem 2 supplements this by analyzing spike pattern changes. It demonstrates that, under adversarial perturbations, SNN activations only change in a limited number of regions, defined by the threshold, due to the piecewise linear nature of SNNs. Both theorems aim to show that reducing the membrane potential distribution of neurons near the threshold enhances adversarial robustness.
>
> ## Weakness 2:I believe the proof of Theorem 3 may be incorrect. According to Appendix E, the flipping probability from 1 to 0 should be expressed as the conditional probability and the same applies to the flipping probability from 0 to 1.
> **A**:Thank you for this correction. In our proof, we partition the transition probability into two cases based on the relationship between the membrane potential and threshold For Case 1, when $ V[t] \geq V_{\mathrm{th}} $; For Case 2, when $ V[t] < V_{\mathrm{th}} $. The transition from spike to non-spike when $V[t] \geq V_{\mathrm{th}}$, and the reverse transition when $V[t] < V_{\mathrm{th}}$. However, we omitted the explicit conditional probability form in the equations, which may have caused this misunderstanding.
> To clarify this, we revise the formula as
> $$
>     P_{1 \rightarrow 0}  \, = \, P(\eta[t] < V_{\mathrm{th}} - V[t] \mid V[t] \geq V_{\mathrm{th}}) \, = \, \Phi\left( \frac{V_{\mathrm{th}} - V[t]}{\sigma} \right),
> $$
>
> $$
>     P_{0 \rightarrow 1}
> \, = \, P(\eta[t] \geq V_{\mathrm{th}} - V[t]  \mid V[t] < V_{\mathrm{th}})
> \, = \, 1 - \Phi\left( \frac{V_{\mathrm{th}} - V[t]}{\sigma} \right).
> $$
> We appreciate your valuable feedback, which improves the rigor of our manuscript.
>
> ## Weakness 3:Theorem 1 is proved under an constraint, whereas the experiments are conducted with perturbations. This inconsistency raises questions about the necessity and practical relevance of the theorem.
> **A**:In the context of adversarial attacks on SNNs, we use the term "gradient sparsity" to describe **the reduction in effective gradient paths available to the attacker**. There are two ways to reduce the effective gradient pathways available to an attacker:(1) directly forcing certain gradients to zero, and (2) decreasing the magnitude of the gradients. Therefore, minimizing the L2 norm satisfies both requirements. optimizing the L2 norm of the Jacobian matrix can:
> - Reduce the overall magnitude of the gradients,
> - Decrease the gradient signal that the attacker can effectively exploit,
> - Thus leading to the sparsification of the "effective gradient paths."
>
> Our TGO module is specifically designed to suppress neuron membrane potentials around the firing threshold, serving as an L2-norm-based constraint objective. Moreover, when the offset $\delta$ exceeds the support of the surrogate gradient and all neuron membrane potentials fall outside the surrogate region, this constraint effectively degenerates into an L0-type penalty on active neurons. Therefore, our analysis and experiments are conducted under an L2-based setting rather than an $\ell_\infty$-norm assumption.

---

> ### Author Response · Authors · 2025-11-21
>
> ## Weakness 4:According to Eq. (7), the constraint loss needs to be computed in a layer-wise manner. Meanwhile, the NLIF module introduces perturbations at every layer. How efficient is TGO in terms of computation compared with other training strategies? Does the training time increase significantly as the network depth grows?
> **A**:We will analyze both the inference and training aspects in terms of computational overhead. TGO-trained SNNs exhibit identical neural dynamics to standard SNNs during inference, without introducing any additional computational cost. Notably, TGO enhances energy efficiency, as its optimization objective encourages neurons to remain further from their firing thresholds, naturally reducing the overall spike firing rate during inference.
>
>
> |model|Clean|FGSM||RFGSM||
> |:-:|:-:|:-:|:-:|:-:|:-:|
> ||Acc.|Acc.|Firing Rate|Acc.|Firing Rate|
> AT| 68.57 | 21.18 | 38.61% | 46.60 | 35.47% |
> |TGO+AT(Ours)| 64.49 | 41.99 | 21.8%|55.01 | 23.7% |
>
> As demonstrated in the above Table, we analyze the average spike firing rates of baseline and TGO-optimized SNNs under various attack scenarios. The results consistently show that TGO-optimized SNNs achieve lower spike firing rates compared to the baseline, indicating superior energy efficiency rather than overhead. Therefore, TGO not only enhances adversarial robustness but also improves energy efficiency, making it particularly valuable for energy-constrained edge deployment scenarios.
>
> During the training process, TGO primarily increases energy consumption in two areas. First, the membrane potential constraint introduces an additional constant calculation. As shown in Equation 7, this does not involve floating-point multiplication but adds approximately $H \times W \times C$ floating-point additions each layer. The additional computation introduced by the membrane potential constraint is $3.2 \times 10^6$, which is approximately 0.9% of the total computation $3.42 \times 10^8$ in each layer on VGG. Therefore, TGO does not introduce more nonlinear complexity and has a negligible impact on the overall training overhead.
> ## Weakness 5:The paper does not specify the value of in Eq. (7) or explain how it was chosen. Please clarify it.
> Eq.7 is used to describe the overall range of membrane potential constraints. We will explain it in detail as follows:
> $$
> \mathcal{C}\bigl(V_l(t)\bigr)
> = \frac{1}{TN} \sum_{i=1}^{N}
> \max\left(0,\ \delta - \left| V_i(t) - V_{\mathrm{th}} \right|\right)
> $$
>
> where $V_{\text{th}}$ is the threshold of the spiking neuron, and $V_i(t) $ represents the membrane potential of the i-th neuron at time $t$ in layer $l$. The hyperparameter $\delta$ establishes a margin around $V{\text{th}}$ where potential values that are close to the threshold incur proportional penalties. In the experiments conducted in this paper, $\delta$ is typically defined as 0.7. This is because most surrogate gradient functions have small gradient values at a distance of 0.7 from $V_{\text{th}}$, which helps ensure that the L2 norm of the SNN remains small.
>
>
> When the membrane potential $V_i(t) $ deviates from the threshold by more than $\delta$ (either above or below), no constraint is imposed on these neurons. However, for neurons whose membrane potential lies within the range $[V_{\text{th}} - \delta, V_{\text{th}} + \delta]$, an additional constraint is applied.
>
> The total constraint strength across all layers' neurons is then computed as $ C(V(t)_l)$. This constraint is treated as a parallel term to the overall optimization loss and is propagated through the backpropagation of gradients, guiding the network to update its synaptic weights accordingly.
>
> ## Weakness 6:The paper contains some typos, though they do not affect my overall rating. A few examples are listed below:
> Thank you very much for pointing out these oversights. We will correct these inaccurate descriptions in the revised manuscript. We sincerely appreciate your constructive guidance on our work.

---

> > ### Comment · Reviewer_FMzT · 2025-11-24
> > **After the rebuttal**
> >
> > Thank you for the response. The replies address most of my concerns. My remaining comments are as follows:
> >
> > 1. Since Theorem 2 is an important extension of Thereom 1, it should be reorganized with simplified and unified notation so that readers can more easily follow the derivation.
> >
> > 2. I do not fully agree with the statement that “gradient sparsity describes the reduction in effective gradient paths available to the attacker,” as the term sparsity is mathematically associated with the $\ell_0$ norm only.  Although I acknowledge the authors’ intuitive explanation, I hope the paper will avoid potential conceptual confusion in the final version.
> >
> > 3. Please revise all inaccurate or unclear parts in the final manuscript, such as those mentioned in W2 and W6.
> >
> > After reading the rebuttal, I have decided to raise my score.

---

> > > ### Author Response · Authors · 2025-11-25
> > >
> > > We are grateful for the insightful comments. We will address all of them in the revised version of our manuscript. We sincerely appreciate your recognition of our work.

---

### Official Review · Reviewer_TQh4 · 2025-10-27

**Soundness:** 2
**Presentation:** 3
**Contribution:** 2
**Rating:** 6
**Confidence:** 4

**Summary:**

This paper proposed a Threshold Guarding Optimization (TGO) method for enhancing the robustness of SNNs. TGO aims to reduce the number of threshold-neighboring spiking neurons, thereby decreasing the state-flipping probability. Noisy-LIF neurons are also adopted to eliminate the influence of adversarial perturbations. Experiments results show that TGO surpasses SOTA SNN-based adversarial defense methods.

**Strengths:**

1.The reasoning of the paper is clear and coherent. Reducing the number of threshold-neighboring spiking neurons provides a new insight in enhancing the robustness of SNNs.

2.The theoretical analysis of the paper is reasonable.

3.The improvement of TGO is significant, improving the robustness effectively. (Only if the experimental results are convincing, see weaknesses below)

**Weaknesses:**

1. In Line 021 in abstract and Line 061 in introduction, the authors mentioned their method can enhance ‘gradient sparsity’. Normally the sparsity corresponds to L0-norm [1]. However, in Theorem 1, the author aims to optimize L2-norm of the Jacobian matrix, which is inconsistent to optimizing the gradient sparsity. The term ‘sparsity’ seems inappropriate.

2. What is Figure 2 used for? The main text does not mention or introduce Figure 2, leaving Figure 2 alone. What is $H[t]$ in Figure 2(a) and Figure 2(d)? Why does the state-flipping probability correspond to $H[t]$ (or $U[t]$) instead of $V[t]$? In Figure 2(c), it seems that your method only penalizes membrane potential under threshold, and membrane potential beyond threshold remains unchanged.

3. The experimental results are unconvincing.
- 3.1 The authors only conducted experiments in CIFAR100 (and a small experiment in DVS-CIFAR10). In Line 335, the authors mentioned CIFAR10, but I cannot see any experiment of CIFAR10 even in Appendix.
- 3.2 The authors only adopted ANN-based attacks. As the paper focuses on SNNs, SNN-based attacks such as [2][3] must be included for comprehensive evaluation.
- 3.3 In Figure 3, your method TGO+AT (about 59%) is lower than SR+AT (about 62%) in clean accuracy. However, in Table 1, TGO+AT (64.49%) is higher than SR+AT (60.37%) in clean accuracy.
- 3.4 Moreover, the adversarial accuracy in APGD and MTPGD in Table 2 is significantly higher than PGD in Table 1. As APGD is much stronger than PGD, why does this situation occur? For instance, in Line 367, PGD10 obtained 22.75% accuracy, while in Line 399, APGD10 only obtained 29.19% accuracy.

References:

[1] Liu, Yujia, et al. "Enhancing Adversarial Robustness in SNNs with Sparse Gradients." International Conference on Machine Learning. PMLR, 2024.

[2] Lun, Li, et al. "Towards Effective and Sparse Adversarial Attack on Spiking Neural Networks via Breaking Invisible Surrogate Gradients." Proceedings of the Computer Vision and Pattern Recognition Conference. 2025.

[3] Hao, Zecheng, et al. "Threaten spiking neural networks through combining rate and temporal information." The Twelfth International Conference on Learning Representations. 2024.

**Questions:**

1. In Line 079, ‘TGO combined with vanilla SNNs surpasses those adversarial training strategies for the first time’. Which experiment supports this contribution?

2. Typos in formulas. Like Theorem 1, Line 272 Vth. Please check the whole manuscript.

---

> ### Author Response · Authors · 2025-11-21
>
> We sincerely appreciate your recognition of the innovative aspects of our proposed TGO method. Below, we provide detailed responses addressing each of your **concerns**.
>
> ## Weakness 1: In Line 021 in abstract and Line 061 in introduction, the authors mentioned their method can enhance ‘gradient sparsity’. Normally the sparsity corresponds to L0-norm. However, in Theorem 1, the author aims to optimize L2-norm of the Jacobian matrix, which is inconsistent to optimizing the gradient sparsity. The term ‘sparsity’ seems inappropriate.
> In the context of adversarial attacks on SNNs, we use the term "gradient sparsity" to describe **the reduction in effective gradient paths available to the attacker**. There are two ways to reduce the effective gradient pathways available to an attacker:(1) directly forcing certain gradients to zero, and (2) decreasing the magnitude of the gradients. Therefore, minimizing the L2 norm satisfies both requirements. optimizing the L2 norm of the Jacobian matrix can:
> - Reduce the overall magnitude of the gradients,
> - Decrease the gradient signal that the attacker can effectively exploit,
> - Thus leading to the sparsification of the "effective gradient paths."
>
> Our TGO method only degenerates into the L0 norm when $\delta$ in Eq. 7 is equal to the surrogate gradient boundary range. However, this significantly impacts the clean accuracy of the SNN. In the revised version, we will more clearly use the term "effective gradient path sparsification" to avoid confusion with traditional L0 sparsity.
>
>
> ## Weakness 2:What is Figure 2 used for? The main text does not mention or introduce Figure 2, leaving Figure 2 alone. What is in Figure 2(a) and Figure 2(d)? Why does the state-flipping probability correspond to $H(t)$(or $U(t)$) instead of $V(t)$? In Figure 2(c), it seems that your method only penalizes membrane potential under threshold, and membrane potential beyond threshold remains unchanged.
> I will address your question from the following three perspectives.
> (1).Figure 2 provides an overview of our TGO method. We apologize for not providing a ()detailed explanation in the main text. Specifically,(a) The TGO method integrates membrane potential constraints with noisy LIF neuron models for adversarial defense. (b) Gradient-based adversarial attacks illustrate the impact of disturbances on input images.(c) The joint optimization of the objective and constraint functions drives neuron membrane potentials away from the firing threshold.(d) The noisy LIF model effectively mitigates the likelihood of state flips caused by small input disturbances, thereby enhancing model stability.
>
> (2). We apologize for the oversight in the variable definitions. We will unify $U$ and $V $ as $ V $ in the updated manuscript. As for $ H $, it represents the postsynaptic membrane potential, distinct from $ V $. Specifically, $H$ refers to the membrane potential of a neuron after a spike reset or decaying membrane potential.
>
> (3).Finally, TGO not only constrains the membrane potential below the threshold but also constrains both sides of the threshold. Specifically, it increases the membrane potential above the threshold while decreasing the membrane potential below the threshold.
> $$
> \mathcal{C}\bigl(V_l(t)\bigr)
> = \frac{1}{TN} \sum_{i=1}^{N}
> \max\left(0,\ \delta - \left| V_i(t) - V_{\mathrm{th}} \right|\right)
> $$
> where $V_{\text{th}}$ is the threshold of the spiking neuron, and $V_i(t) $ represents the membrane potential of the i-th neuron at time $t$ in layer $l$. The hyperparameter $\delta$ establishes a margin around $V{\text{th}}$ where potential values that are close to the threshold incur proportional penalties. In the experiments conducted in this paper, $\delta$ is typically defined as 0.7. This is because most surrogate gradient functions have small gradient values at a distance of 0.7 from $V_{\text{th}}$, which helps ensure that the L2 norm of the SNN remains small. When the membrane potential $V_i(t) $ deviates from the threshold by more than $\delta$ (either above or below), no constraint is imposed on these neurons. However, for neurons whose membrane potential lies within the range $[V_{\text{th}} - \delta, V_{\text{th}} + \delta]$, an additional constraint is applied.

---

> ### Author Response · Authors · 2025-11-21
>
> ## Weakness 3:The experimental results are unconvincing.
> ### Weakness 3.1:The authors only conducted experiments in CIFAR100 (and a small experiment in DVS-CIFAR10). In Line 335, the authors mentioned CIFAR10, but I cannot see any experiment of CIFAR10 even in Appendix.
>  We apologize for not presenting the CRIAF10 experimental results due to space limitations. We believe that the CRIAF100 data better demonstrates the generalizability and reliability of our method. However, we did conduct comparison experiments with CRIAF10, and the results are as follows.
>
> | **Train**  | **Method**  | **Clean** | **FGSM** | **RFGSM** | **PGD7** | **PGD10** | **PGD20** | **PGD40** |
> |------------|-------------|-----------|----------|-----------|----------|-----------|-----------|-----------|
> | BPTT   | Vanilla[1] | 93.32 | 14.05 | 31.21 | 0.00 | 0.00 | 0.00 | 0.00 |
> |    BPTT        | TGO (Ours) | 88.79 | **51.40** | **71.38** | **11.67** | **6.14** | **1.52** | **0.45** |
> | AT     | AT | **91.32** | 39.14 | 74.31 | 21.15 | 17.45 | 14.41 | 12.93 |
> |      AT      | TGO (Ours) | 88.16 | **63.03** | **79.69** | **42.52** | **35.01** | **24.76** | **20.11** |
> | RAT    | RAT | **91.44** | 42.02 | 75.89 | 23.90 | 19.81 | 16.24 | 14.18 |
> |      RAT      | TGO (Ours) | 87.33 | **69.16** | **79.28** | **54.59** | **47.69** | **38.07** | **33.13** |
>
> In the revised version, we will include the CRIAF10 experimental results in the appendix. We appreciate you pointing out this oversight.
>
> ### Weakness 3.2:The authors only adopted ANN-based attacks. As the paper focuses on SNNs, SNN-based attacks such as [2][3] must be included for comprehensive evaluation.
>
> A: Following your suggestion, we conducted additional experiments on SNN-specific attacks. Due to time constraints, we first tested the attack from HART[3], with results as follows:
>
> | **Method**    | **HART-fgsm** | **HART-pgd10** |
> |---------------|---------------|----------------|
> | AT       | 20.65         | 9.35           |
> | AT+TGO    | 45.65         | 30.21          |
>
> The experiments demonstrate that TGO significantly enhances the adversarial robustness of SNNs. We will continue testing the results from PDSG [2] in future work. Thank you for your insightful feedback, which has greatly improved the quality of our manuscript.
>
> ### Weakness 3.3: In Figure 3, your method TGO+AT (about 59%) is lower than SR+AT (about 62%) in clean accuracy. However, in Table 1, TGO+AT (64.49%) is higher than SR+AT (60.37%) in clean accuracy.
>
> We apologize for the misunderstanding, which resulted from an oversight on our part. In our TGO method, there is a hyperparameter $\lambda$ that controls the strength of the membrane potential constraint. As shown bolew:
> $$
>     \mathcal{L}(\mathbf{x}, \lambda) = \mathcal{L}oss(\mathbf{x}) + \lambda \sum_l \mathcal{C}(V(t)_l).
> $$
> A larger $\lambda$ increases adversarial robustness but reduces clean accuracy, making the choice of $\lambda$ a trade-off. we performed experiments on CIFAR-100 using the WRN-16 architecture. The results are shown below.
> |model|$\lambda_{max}$|Clean|FGSM|RFGSM|PGD7|PGD10|PGD20|PGD30|PGD40|PGD50|
> |:-:|:-:|:-:|:-:|:-:|:-:|:-:|:-:|:-:|:-:|:-:|
> |TGO+AT| 0.2  | 66.93 | 34.53 | 53.34 | 18.93 | 14.32 | 9.71 | 8.00 | 7.50 | 7.14 |
> |TGO+AT| 0.3  | 65.80 | 39.02 | 54.32 | 24.23 | 19.35 | 13.58 | 11.92 | 10.74 | 10.50 |
> |TGO+AT| 0.4  | 64.49| 41.99 | 55.01 | 27.78 | 22.75 | 15.99 | 14.59 | 14.12 | 13.79 |
> |TGO+AT| 0.6  |59.56| 48.27 | 54.37 | 41.08 | 36.65 | 31.15 | 29.19 | 28.46 | 28.12 |
>
> In Table 1, the TGO method uses $\lambda = 0.4$, which leads to higher clean accuracy compared to SR+AT. However, in Figure 3, where we test different attack radii, $\lambda = 0.6$ is used, resulting in lower clean accuracy than TGO+AT. We will clarify the choice of $\lambda$ in the revised version. Thank you again for pointing out this oversight.
>
> ### Weakness 3.4: Moreover, the adversarial accuracy in APGD and MTPGD in Table 2 is significantly higher than PGD in Table 1. As APGD is much stronger than PGD, why does this situation occur? For instance, in Line 367, PGD10 obtained 22.75% accuracy, while in Line 399, APGD10 only obtained 29.19% accuracy.
> This issue is similar to the previous one. In our experiments, the hyperparameter $\lambda$ chosen for each table is different. In Table 1, the TGO method uses $\lambda = 0.4$. In Table 2, due to the higher attack strength of APGD, we deliberately chose a higher $\lambda=0.6$, sacrificing more clean accuracy for greater adversarial robustness. This explains why TGO performs better than PGD under the APGD attack, as you pointed out. In the revised manuscript, we will clearly label the chosen value of $\lambda$ in the table headers.

---

> ### Author Response · Authors · 2025-11-21
>
> ## Questions 1: In Line 079, ‘TGO combined with vanilla SNNs surpasses those adversarial training strategies for the first time’. Which experiment supports this contribution?
> We apologize for the lack of clarity in our expression. We intended to say that under FGSM attacks, TGO performs similarly to AT on VGG-11 and slightly better on WRN-16. However, this comparison was imprecise, as TGO is outperformed by these strategies under other attacks(PGD, APGD). Notably, TGO can be effectively combined with these methods to further enhance adversarial robustness. We will revise this to state that TGO can effectively complement other adversarial training strategies to improve the robustness of SNNs, rather than comparing their performance directly.
>
>
> ## Reference:
> [1] Deng S, Li Y, Zhang S, et al. Temporal efficient training of spiking neural network via gradient re-weighting[J]. ICLR, 2022.
>
> [2] Lun, Li, et al. "Towards Effective and Sparse Adversarial Attack on Spiking Neural Networks via Breaking Invisible Surrogate Gradients." Proceedings of the Computer Vision and Pattern Recognition Conference. 2025.
>
> [3] Hao, Zecheng, et al. "Threaten spiking neural networks through combining rate and temporal information." The Twelfth International Conference on Learning Representations. 2024.

---

### Official Review · Reviewer_3pNN · 2025-10-30

**Soundness:** 3
**Presentation:** 3
**Contribution:** 2
**Rating:** 6
**Confidence:** 5

**Summary:**

This paper introduces a Threshold Guarding Optimization (TGO) method to enhance the adversarial robustness of SNNs. By regulating neuron membrane potentials and employing probabilistic firing via noisy neurons, TGO significantly reduces vulnerability to adversarial perturbations, outperforming existing methods in various adversarial scenarios.

**Strengths:**

- This paper is well-written and logically structured, making complex concepts accessible and easy to follow.
- This paper provides a mathematical analysis linking “threshold-neighboring neurons” to adversarial vulnerability, which is a novel and interesting framework for SNN robustness research.
- The authors demonstrate the effectiveness of the proposed TGO method across a wide range of adversarial attack scenarios. The experiments span multiple datasets, network architectures, and adversarial settings, showing strong and consistent results, including outperforming SOTA baselines.

**Weaknesses:**

- The method introduces additional hyper-parameters such as coefficient parameter $\lambda$ and noise level $\sigma$. However, the effectiveness of the $\lambda$ scheduling and sensitivity of the noise level $\sigma$ is missing.
- The paper does not report the additional training cost introduced by the TGO compared to baselines such as adversarial trainings (AT, RAT).

**Limitation**

According to the reported results, the proposed method appears to reduce clean accuracy, indicating a potential trade-off between robustness and standard performance.

**Questions:**

Can the authors provide a more detailed derivation or intuition for Eq. (7)?

---

> ### Author Response · Authors · 2025-11-21
>
> We sincerely appreciate your recognition of the innovative aspects of our proposed TGO method. Below, we provide detailed responses addressing each of your **concerns**.
>
> ## Weakness 1: The method introduces additional hyper-parameters such as coefficient parameter $\lambda$ and noise level  $\theta$. However, the effectiveness of the scheduling and sensitivity of the noise level is missing.
> Thank you for this valuable comment. We have conducted additional experiments to investigate both $\lambda$ and noise level $\theta$ in more detail. For the constraint constant $\lambda$, we performed experiments on CIFAR-100 using the WRN-16 architecture. The results are shown below.
> |model|$\lambda$|Clean|FGSM|RFGSM|PGD7|PGD10|PGD20|PGD30|PGD40|PGD50|
> |:-:|:-:|:-:|:-:|:-:|:-:|:-:|:-:|:-:|:-:|:-:|
> |TGO+AT| 0.2  | 66.93 | 34.53 | 53.34 | 18.93 | 14.32 | 9.71 | 8.00 | 7.50 | 7.14 |
> |TGO+AT| 0.3  | 65.80 | 39.02 | 54.32 | 24.23 | 19.35 | 13.58 | 11.92 | 10.74 | 10.50 |
> |TGO+AT| 0.4  | 64.05| 41.99 | 55.01 | 27.78 | 22.75 | 15.99 | 14.59 | 14.12 | 13.79 |
> |TGO+AT| 0.6  |61.49| 48.27 | 54.37 | 41.08 | 36.65 | 31.15 | 29.19 | 28.46 | 28.12 |
>
> Experimental results show that a larger $\lambda$ leads to stronger adversarial robustness, but at the cost of lower clean accuracy. Therefore, $\lambda$ should be chosen appropriately according to the target application to balance robustness and accuracy. In addition, we conduct a detailed study of the noise level  $\theta$ , and the corresponding results are reported below.
> |model|$\theta$|Clean|FGSM|RFGSM|
> |:-:|:-:|:-:|:-:|:-:|
> |TGO| 0  | 66.93 | 17.12 | 25.87 |
> |TGO| 0.2  | 64.80 |17.26 | 29.45 |
> |TGO| 0.4  | 63.05| 20.46| 36.06|
> |TGO| 0.6  | 58.05| 21.21| 37.21|
>
> The experimental results indicate that, similar to $\lambda$, $\theta$ significantly reduces clean accuracy. Furthermore, when ($\theta$ = 0.6), the improvement in robustness is not substantial. Therefore, we selected $\theta$ with a mean of 0 and a variance of 0.4 for all experiments. We appreciate your constructive feedback and guidance.
>
> ## Weakness 2:Analyze the computational resource and time overhead introduced by this requirement.
> **A**:We will analyze both the inference and training aspects in terms of computational overhead. TGO-trained SNNs exhibit identical neural dynamics to standard SNNs during inference, without introducing any additional computational cost. Notably, TGO enhances energy efficiency, as its optimization objective encourages neurons to remain further from their firing thresholds, naturally reducing the overall spike firing rate during inference.
>
> |model|Clean|FGSM||RFGSM||
> |:-:|:-:|:-:|:-:|:-:|:-:|
> ||Acc.|Acc.|Firing Rate|Acc.|Firing Rate|
> AT| 68.57 | 21.18 | 38.61% | 46.60 | 35.47% |
> |TGO+AT(Ours)| 64.49 | 41.99 | 21.8%|55.01 | 23.7% |
>
> As demonstrated in the above Table, we analyze the average spike firing rates of baseline and TGO-optimized SNNs under various attack scenarios. The results consistently show that TGO-optimized SNNs achieve lower spike firing rates compared to the baseline, indicating superior energy efficiency rather than overhead. Therefore, TGO not only enhances adversarial robustness but also improves energy efficiency, making it particularly valuable for energy-constrained edge deployment scenarios.
>
>
>
> During the training process, TGO primarily increases energy consumption in two areas. First, the membrane potential constraint introduces an additional constant calculation. As shown in Equation 7, this does not involve floating-point multiplication but adds approximately $H \times W \times C$ floating-point additions. The additional computation introduced by the membrane potential constraint is $3.2 \times 10^6$, which is approximately 0.9% of the total computation $3.42 \times 10^8$ in VGG （AT/RAT）. Therefore, TGO does not introduce more nonlinear complexity and has a negligible impact on the overall training overhead.

---

> ### Author Response · Authors · 2025-11-21
>
> ## Questions 1: Can the authors provide a more detailed derivation or intuition for Eq. (7). Eq.7 is used to describe the overall range of membrane potential constraints. We will explain it in detail as follows:
> $$
> \mathcal{C}\bigl(V_l(t)\bigr)
> = \frac{1}{TN} \sum_{i=1}^{N}
> \max\left(0,\ \delta - \left| V_i(t) - V_{\mathrm{th}} \right|\right)
> $$
>
> where $V_{\text{th}}$ is the threshold of the spiking neuron, and $V_i(t) $ represents the membrane potential of the i-th neuron at time $t$ in layer $l$. The hyperparameter $\delta$ establishes a margin around $V{\text{th}}$ where potential values that are close to the threshold incur proportional penalties. In the experiments conducted in this paper, $\delta$ is typically defined as 0.7. This is because most surrogate gradient functions have small gradient values at a distance of 0.7 from $V_{\text{th}}$, which helps ensure that the L2 norm of the SNN remains small.
>
>
> When the membrane potential $V_i(t) $ deviates from the threshold by more than $\delta$ (either above or below), no constraint is imposed on these neurons. However, for neurons whose membrane potential lies within the range $[V_{\text{th}} - \delta, V_{\text{th}} + \delta]$, an additional constraint is applied.
>
> The total constraint strength across all layers' neurons is then computed as $ C(V(t)_l)$. This constraint is treated as a parallel term to the overall optimization loss and is propagated through the backpropagation of gradients, guiding the network to update its synaptic weights accordingly.

---

### Author Response · Authors · 2025-12-03
**Final Remarks**

We sincerely thank all reviewers for their constructive feedback, which has significantly improved our manuscript. We are grateful for the reviewers' recognition of our work, particularly in the following aspects:

1. Clear writing, rigorous logic, and well-organized structure (3pNN, TQh4, FMzT, 23Ay)
2. Sufficient experiments validating the effectiveness of the proposed method (3pNN, TQh4, FMzT)
3. Theoretically grounded, flexible, easy-to-understand, and practical methodology (3pNN, TQh4, FMzT, 23Ay)

Based on reviewer feedback, the primary concerns and our corresponding revisions are as follows:

1. Hyperparameter λ Specification（3pNN, TQh4, FMzT, 23Ay）: To eliminate ambiguity in the selection of the regularization coefficient λ, we have explicitly annotated the hyperparameter values in all table captions and figure legends.

2. Sparsity Terminology（3pNN, TQh4, FMzT）: To prevent potential misinterpretation, we have revised all references to "sparsity" to explicitly denote gradient path sparsity, providing a more precise characterization of the proposed regularization mechanism.

3. Computational Overhead of TGO（3pNN, FMzT, 23Ay）: We clarify that TGO incurs no additional inference overhead, as it remains fully consistent with standard SNN forward propagation. Moreover, the membrane potential constraint imposed by TGO yields lower spike firing rates, thereby enhancing energy efficiency on neuromorphic hardware.

Reviewers 3pNN and TQh4 expressed a clearly positive attitude toward our work in their initial reviews. Notably, **Reviewer FMzT, who initially raised concerns (W2–W5), acknowledged our explanations during the first rebuttal round and revised their score from negative to 6, indicating acceptance of our work**. As for Reviewer 23Ay, we have provided thorough and point-by-point responses to all concerns raised; however, despite our efforts, **no further discussion or feedback was received from this reviewer throughout the rebuttal period.**

In summary, we sincerely appreciate the efforts and valuable suggestions from all reviewers and have incorporated their feedback into this revised manuscript. We also extend our gratitude to the Area Chair for handling our submission.

---

### Meta-Review · Area_Chair_B88Z · 2026-01-06

**Summary:**

This paper proposes Threshold Guarding Optimization (TGO), a training approach for improving the adversarial robustness of spiking neural networks by discouraging neurons from operating near firing thresholds and by introducing probabilistic firing behavior.
The main concerns raised is the rigor and clarity of the theoretical analysis, the completeness and consistency of the experimental evaluation, and the interpretation of robustness claims in the presence of stochastic inference. In particular, one reviewer questioned whether the adversarial evaluation protocol was sufficiently rigorous, given the use of randomness, the limited use of EoT and adaptive attacks in the main experiments, and inconsistencies across tables and hyperparameter settings. Other reviewers raised more localized issues, including missing hyperparameter details, unclear figures and terminology, lack of SNN-specific attacks, and limited discussion of computational overhead.

The rebuttal was detailed and addressed many of these points through additional experiments, clarifications, and corrections. Several reviewers indicated that their concerns were largely resolved.

**Reviewer Concerns:**

### Concerns addressed by the rebuttal:

- The authors clarified the choice and sensitivity of key hyperparameters and added ablation studies to illustrate the robustness–accuracy trade-off.

- Computational overhead was analyzed, with explanations showing negligible training cost increases and no inference-time overhead, alongside reduced spike firing rates.

- The terminology around “gradient sparsity” was clarified and revised to reduce potential confusion.

- The experimental evaluation was expanded to include CIFAR-10 results, SNN-specific attacks, surrogate-gradient variants, ensemble attacks, EoT evaluations, and additional datasets and architectures.

- Several theoretical and presentation issues were corrected, including errors in proofs, unclear notation, and inconsistencies between tables and figures.

### Concerns that remain outstanding:

- Evaluations on EoT, random restarts, and adaptive attack protocols.

- The huge drop of the natural accuracy.

- While many details were clarified, the overall presentation remains dense, and some parts of the paper may still be difficult to follow.

**Reviewer Scores:**

Reviewer 3pNN: Likely to remain at 6 (weak accept) after the rebuttal.

Reviewer TQh4: Likely to remain at 6, given the additional experiments and clarifications.

Reviewer FMzT: Initially 4; explicitly stated that concerns were largely addressed and raised the score to 6.

Reviewer 23Ay: Initially 2, after point-by-point responses, the score may raise to 4 or 6.

---

### Decision · Program_Chairs · 2026-01-26

Accept (Poster)